# Modeling the effect of Ross Ice Shelf melting on the Southern Ocean in quasi-equilibrium

Xiying Liu[*]

Institute of Atmospheric Physics, Chinese Academy of Sciences, Beijing, 100029, China

*Correspondence to Xiying Liu (liuxy@lasg.iap.ac.cn)*

**Abstract.** To study the influence of basal melting of the Ross Ice Shelf (BMRIS) on the Southern Ocean (ocean southward of 35˚ S) in quasi-equilibrium, numerical experiments with and without the BMRIS effect were performed using a global ocean-sea ice-ice shelf coupled model. In both experiments, the model started from a state of quasi-equilibrium ocean and

was integrated for 500 years forced by CORE (Coordinated Ocean-ice Reference Experiment) normal year atmospheric fields. The simulation results of the last 100 years were analyzed. The melt rate averaged over the entire Ross Ice Shelf is 0.25 m a$^{-1}$, which is associated with a freshwater flux of 3.15 mSv (1 mSv = $10^3$ m$^3$ s$^{-1}$). The extra freshwater flux decreases the salinity in the region from 1500 m depth to the sea floor in the southern Pacific and Indian Oceans, with a maximum difference of nearly 0.005 PSU in the Pacific Ocean. Conversely, the effect of concurrent heat flux is mainly confined to the

middle depth layer (approximately 1500 m to 3000 m). The decreased density due to the BMRIS effect, together with the influence of ocean topography, creates local differences in circulation in the Ross Sea and nearby waters. Through advection by the Antarctic Circumpolar Current, the flux difference from BMRIS gives rise to an increase of sea ice thickness and sea ice concentration in the Ross Sea adjacent to the coast and ocean water to the east. Warm advection and accumulation of warm water associated with differences in local circulation decrease sea ice concentration on the margins of sea ice cover

adjacent to open water in the Ross Sea in September. The decreased water density weakens the sub-polar cell as well as the lower cell in the global residual meridional overturning circulation. Moreover, we observe accompanying reduced southward meridional heat transport at most latitudes of the Southern Ocean.

---

[*] Current affiliation: College of Oceanography, Hohai University, Nanjing, 210098, China.

# 1 Introduction

Ice shelf melting, which accounts for 55% of ice mass loss from Antarctica, is one of the main sources of freshwater to the Antarctic coastal ocean (Mathiot et al.,2017). Ice accumulated on ice sheets is mostly lost to the oceans by melting underneath the ice shelves or by calving of icebergs. Floating ice shelves around Antarctica are thinning substantially, driven primarily by melting at the ice-ocean interface ( Rignot et al., 2013; Paolo et al., 2015). Ice shelf melting exceeds the calving flux (Rignot et al., 2013; Depoorter et al., 2013) and contributes significantly to the fresh water balance in ice shelf areas around Antarctica (Beckmann and Goosse, 2003). The circulation that occurs in sub-ice-shelf cavities is markedly different from that in the open ocean, consisting largely of thermohaline circulation forced by melting and freezing processes at the ice shelf base. This circulation is of more than local importance because it plays a key role in the production of Antarctic bottom water (AABW), a driver of global thermohaline circulation (Walker and Holland, 2007). The sub-ice freshwater input has various implications for the Southern Ocean. These are most pronounced in the Weddell and Ross Seas where large caverns are connected to broad continental shelves (Hellmer, 2004). Mass exchange between the Antarctic Ice Sheet and the Southern Ocean has drawn substantial research attention (Rowley et al., 2007; Kusahara and Hasumi, 2013).

Basal melting of the ice shelves has long been of interest because of its importance to the mass balance of the Antarctic ice sheet (Nost and Foldvik, 1994). The amount of basal melting from ice shelves has been estimated by many studies (for example, Hellmer and Jacobs (1995); Rignot et al. (2013); Moholdt et al. (2014); and others). Ice shelf basal melting is approximately 0.85 m a$^{-1}$ over the circumpolar continental shelf area (Rignot et al., 2013), exceeding P–E (Precipitation minus Evaporation) by a factor of at least two (Beckmann and Goosse, 2003). Because injection of this freshwater occurs at depth rather than at the ocean surface, it has a different impact on the stability of the coastal ocean than P–E forcing. The quantification of basal mass loss under changing climate conditions is important for projections of the dynamics of Antarctic ice streams and ice shelves, as well as global sea level rise (Hellmer et al., 2012).

The need for numerical modeling of ice shelf–ocean interactions is particularly acute due to a lack of extensive observational data, which results from the physical inaccessibility of the areas of interest. Besides, it is difficult to infer sub-ice-shelf circulation from borehole observations, creating a significant need for numerical models (Walker and Holland, 2007; Dinniman et al.,2016). As illustrated in Table 1, in ice shelf-sea ice-ocean coupled modeling, researchers use different types of ice shelf representations, such as dynamic ice-shelf geometry permitting two-dimensional flow (Grosfeld and Sandhager, 2004), simplified and computationally inexpensive representations that are nevertheless capable of handling significant changes to the shape of the sub-ice-shelf cavity as the shelf profile evolves (Walker and Holland, 2007), thermodynamics with fixed cavity techniques (Losch, 2008; Timmermann et al., 2012), and parameterized schemes for the interaction between ice shelves and the adjacent ocean (Beckmann and Goosse, 2003). The models are mostly circumpolar (Hellmer, 2004; Kusahara and Hasumi, 2013; Mathiot et al., 2017), regional (Galton-Fenzi et al., 2012), or two-dimensional in the yz-plane (Walker et al., 2009).

A few global models were also used for numerical studies. For example, Beckmann and Goosse (2003) studied the ice shelf basal melting effect using a global ocean-sea ice coupled model with a first order parameterization of ice shelf-ocean interaction. Losch (2008) introduced ice shelves into the Massachusetts Institute of Technology general circulation model (MITgcm) and conducted ISOMIP (Ice Shelf–Ocean Model Intercomparison Project) experiments and nearly global (excluding the Arctic Ocean) ocean circulation experiments. In these experiments, results with and without explicit modeling of ice shelf cavities were presented and the analysis was mainly focused on the Weddell Sea and circulation in the Filchner-Ronne Ice Shelf cavity. Timmermann et al. (2012) presented results of ice shelf basal mass loss from a global sea ice-ice shelf-ocean model based on the finite element method, in which the model was forced with daily data from the NCEP/NCAR reanalysis for the period 1958–2010. There are also numerous other recent modeling studies on ice shelves that employed regional, circumpolar, or global models; Asay-Davis et al. (2017) provided a thorough review of these studies. However, to research the effect of ice-shelf melting on the ocean in quasi-equilibrium, it is necessary to use a global model with thermodynamically active ice-shelf cavities and perform integration over hundreds of years. This type of research has not previously been conducted.

The Antarctica possesses the majority of the world's ice shelves, of which the Ross Ice Shelf (RIS) has the largest area. Nost and Foldvik (1994) studied the circulation under the RIS with a simple analytical model. Using a two-dimensional channel flow model forced by thermohaline differences between the open boundaries and the interior cavity, Hellmer and Jacobs (1995) studied the flow under the RIS and estimated an ice shelf base loss rate of 18–27 cm a$^{-1}$. By comparing model estimates of oceanic CFC-12 concentrations along an ice shelf edge transect to field data collected during three cruises spanning 16 years, Reddy et al. (2010) estimated that the residence time of water in the RIS cavity is approximately 2.2 years and that basal melt rates for the ice shelf average 0.1 m a$^{-1}$. Arzeno et al. (2014) used data from two moorings deployed through RIS, ~6 and ~16 km south of the ice front east of Ross Island, and numerical models to show how the basal melting rate near the ice front depends on sub-ice-shelf ocean variability. However, these studies do not deepen our understanding of the influences of RIS on the Southern Ocean in quasi-equilibrium because the domains of the models employed were not sufficiently large and modeling results were significantly impacted by boundary conditions.

The marginal Ross Sea is an area of deep and bottom water formation. Approximately 25% of the total production rate of AABW comes from the Ross Sea, and basal melting of the ice shelves modifies the characteristics of water masses during the processes of AABW production along the Antarctic continental shelves (Budillon et al, 2011). Antarctic Bottom Water is distinctly colder and fresher than North Atlantic Deep Water and flows northward underneath it in the Atlantic at depths below 4000 m. In this study, we aim to estimate the effect of BMRIS on the Southern Ocean in quasi-equilibrium using a global ice shelf-sea ice-ocean coupled model. The model represents ice shelf-ocean interaction by assuming the RIS to be in a steady state, interacting with the ocean only through thermodynamics.

## 2 Model, datasets and experimental set up

MITgcm (Marshall et al., 1997) is used to carry out the numerical experiments and an Antarctic cavity geometry dataset (Timmermann et al., 2010) is used to obtain the RIS draft. CORE (Coordinated Ocean-ice Reference Experiment) normal year data (Large and Yeager, 2009) are used for atmospheric forcing fields. The MITgcm consists of packages such as atmosphere, ocean, sea ice, and ice shelf for flexible configuration. The parameterizations used in this study include the Gent-McWilliams-Redi eddy parameterization (Redi, 1982; Gent and McWilliams, 1990) and the non-local K-profile vertical mixing parameterization (Large et al., 1994). A sea ice model package with zero-layer thermodynamics (Hibler, 1980) and viscous-plastic rheology (Zhang and Hibler, 1997) is employed. A package of ice shelf thermodynamics (Losch, 2008) named '*shelfice*' is ready for use in the MITgcm[1].

Two experiments are implemented, one with RIS basal melting and one without, denoted as EI and EN, respectively. In both experiments, bathymetry of the RIS cavity is included. Both experiments start from a model restart state of one integration over 1000 years (Liu and Liu, 2012). To improve the vertical resolution of the RIS, that of the upper 1000 m is increased and that below 1500 m is coarsened, whilst maintaining the number of model layers at 30. The layer thicknesses are 10 (x 2), 15, 21, 28, 36, 45, 50 (x 13), 100, 200, 300, 400, 500, 600, 700, and 800 (x 3) m. The current vertical discretization meets the minimum vertical resolution required to resolve ice shelf-ocean processes, where the layer thickness is 100 m (Losch, 2008). The vertical resolution near the bottom is poor; however, this problem is somewhat alleviated by the partial cell treatment of topography (Adcroft et al., 1997). If the RIS cavity is too thin to be resolved by the vertical grids, it will be set to zero. An experiment with RIS excluded from the domain, denoted as EX, is also implemented. The experiment shares the configuration of EN except that the area covered with RIS is treated as land. After preliminary analysis of simulation results, it is realized that the sub-ice-shelf bathymetry gives significant contribution to the differences between the results from EX and EN. The difference in geometry changes local circulation and mixing and leads to changes of overall results compared to or even greater than that in basal melting under the RIS. Under such conditions, it would be difficult to discern the effect of basal melting under the Ross Ice Shelf. Hence the results from EX will not be discussed in later analysis.

Vertical interpolation is used to obtain the model initial fields. Because the RIS was treated as land in the former integration covering more than 1000 years, the initial ocean states in the RIS cavity of the experiments are derived from extrapolation. Cubed sphere grids are used and the horizontal resolution is approximately 150 km. In the original dataset of Antarctic cavity geometry (Timmermann et al., 2010), the area of the cavity is 502024.1 km$^2$; however, in the model, it is only 476924.2 km$^2$ due to the coarse model resolution. This would lead to a reduced freshwater flux. Except for vertical layer division and the *shelfice* package, model parameters used here are identical to those in Liu and Liu (2012). The major

---

[1] For more information on the MITgcm see the latest online document on the MIT website:

http://mitgcm.org/public/r2_manual/latest/online_documents /manual.html.

parameters for the *shelfice* package used in EI are given in Table 2. The model has been integrated for 500 years for each configuration.

Under the current configuration, the whole model domain in the horizontal consists of six cubed sphere faces with Antarctica situated on the 6th face. The ocean bathymetry around Antarctica and the cavity geometry of RIS are shown in Fig. 1. There are 64 x 64 grid cells on each cubed sphere face and the maximum depth of the Southern Ocean is over 6000 m (Fig. 1a). A total of 19 grid boxes are covered by the ice shelf, of which 15 have non-zero cavity thickness and include basal melt calculations in the model. The water-column thickness of the RIS cavity ranges from 50 m to 500 m (Fig. 1b). For the four grid boxes whose ice shelf cavity are removed, the thicknesses of the water columns are less than 42 m, which cannot be resolved with vertical grid cells 50 m in size (from the 8th layer, which is approximately 200 m below the sea surface, the vertical grid size is 50 m). The cavity thickness in the model may be smaller than that in the original dataset with a maximum difference of less than 50 m.

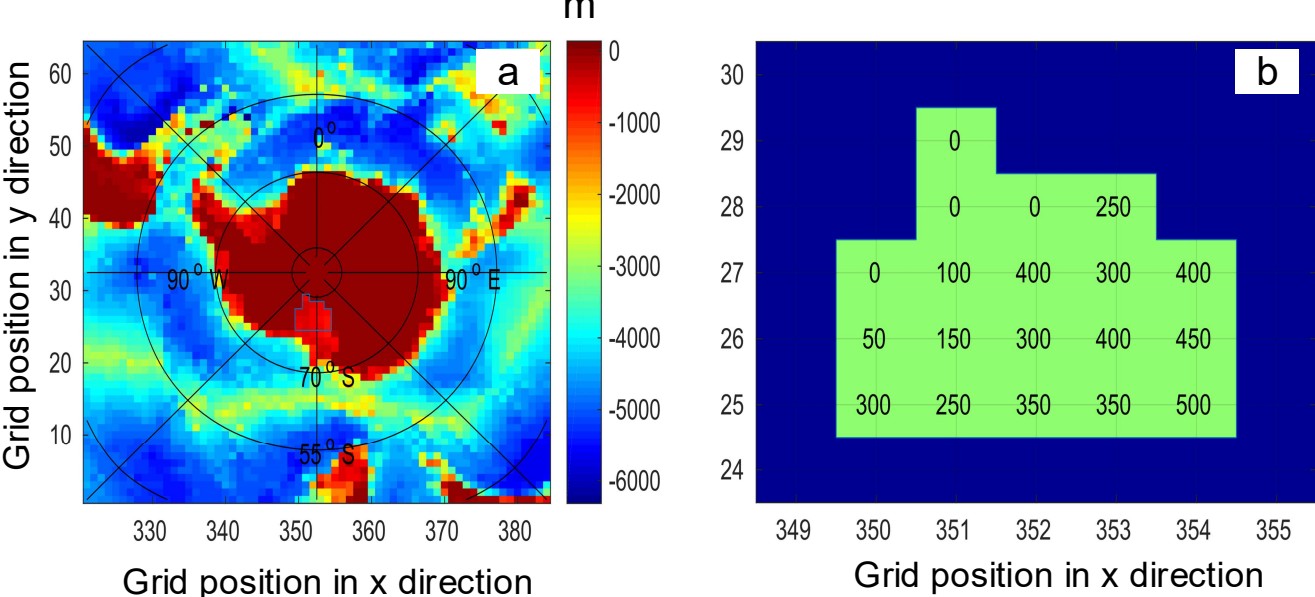

**Fig. 1. (a) Bathymetry of the 6th cubed sphere face in the experiments and (b) cavity geometry of RIS in EI. The numbers on the axes indicate the positions of grids on the model domain. Grid boxes shaded light green in (b) indicate the locations covered by RIS in the model and the numbers in (b) indicate the thickness of the water column in the cavity. The units of bathymetry and water column thickness in the cavity are in m.**

## 3 Results

### 3.1 Simulated basal melting of RIS and its local effects

There is significant interdecadal variability in the simulated basal melt rate, which is smaller in the last 100 years than in other periods (Fig. 2a). This difference in the interdecadal variability may be due to the influence of ocean system adjustment processes. After approximately 400 years of adjustment, the ocean reaches a quasi-equilibrium state and the decadal variability becomes smaller. Therefore, in this study, only integration data of the last 100 years is used. In the simulation result, only 2 out of 15 grid boxes of RIS experience annual mean freezing and the largest melt occurs near the ice shelf front (Fig. 2b). As illustrated in Table 3, the melt rate averaged over the entire RIS is 0.25 m a$^{-1}$, which is comparable to the results of Hellmer and Jacobs (1995), Assmann et al. (2003), Arzeno et al. (2014), and Mathiot et al. (2017), but higher than that of Shabtaie and Bentley (1987), Holland et al. (2003), Dinniman et al. (2007, 2011), Depoorter et al. (2013), and Moholdt et al. (2014), and lower than that of Timmermann et al. (2012). There is no clear difference in the basal melt rate between Ross East and Ross West, different from the result revealed by Rignot et al. (2013). The highest melt occurs in April (approximately 0.269 m a$^{-1}$) and the lowest melt occurs in November (approximately 0.238 m a$^{-1}$) (Fig. 2c). This differs from the results of Holland et al. (2003), in which the largest basal melt rate occurs in November. This difference in seasonality may be due to the combined effects of such factors as melting mechanisms in the modeled ice shelf, adjustment processes of the model system, atmospheric forcing, and the influence of boundary conditions. The modeling system used by Holland et al. (2003) did not incorporate wind and sea ice and the surface temperature and salinity was restored. In a two-year simulation after six years of spin up using a regional model, Dinniman et al. (2007) obtained a maximum basal melt rate in February and a minimum value in September. It should be noted that the phase of seasonal cycle of basal melt is not uniform geographically. For grids adjacent to the ground and far from the ice front, the basal melt is influenced by flux from vertical heat conduction to a greater extent and reflects the air temperature variations much more (figures not shown).

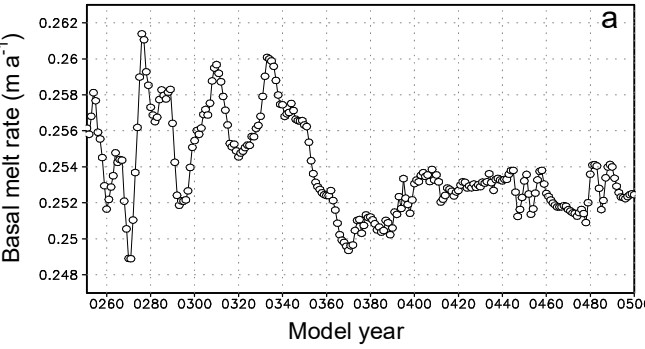

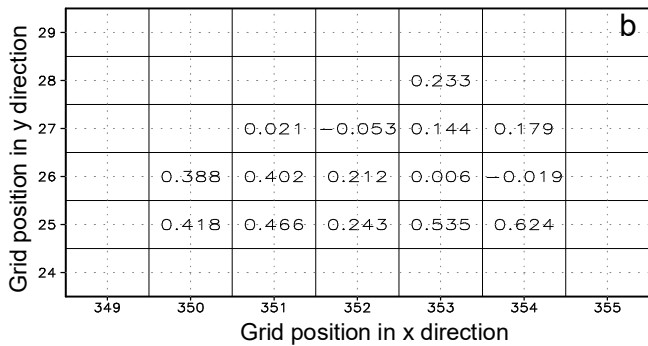

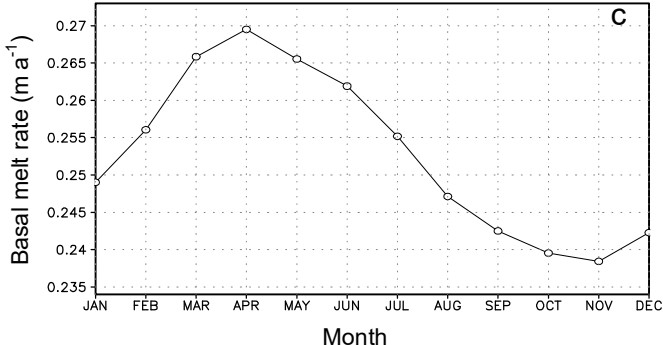

**Fig. 2. Basal melting rates of RIS (m a⁻¹) in EI. (a) Variation of the annual and areal mean melting rate over the last 250 years. (b) Spatial distribution of the mean melting rate over the last 100 years. (c) Seasonal cycle averaged over the ice-shelf area and the last 100 years.**

Due to the inclusion of ice shelf melting in EI, cold and fresh water is supplemented into the RIS cavity and the water there becomes colder and fresher than its counterpart in EN (Fig. 3). The relationship between salinity differences and temperature differences of RIS cavity water for the two experiments is quasi-linear, implying that larger salinity differences

10    correspond to larger temperature differences. This is fundamentally governed by the latent heat formula in the model equations. The maximum decrease of water temperature and salinity in the cavity can reach 0.50 °C and 0.25 PSU, respectively, due to the RIS melting effect and there seems to be no significant difference in feature between inflow anomaly and outflow anomaly in the cavity (Fig. 3). This implies that, in quasi-equilibrium, a significant amount of outflowing freshwater recirculates into the cavity.

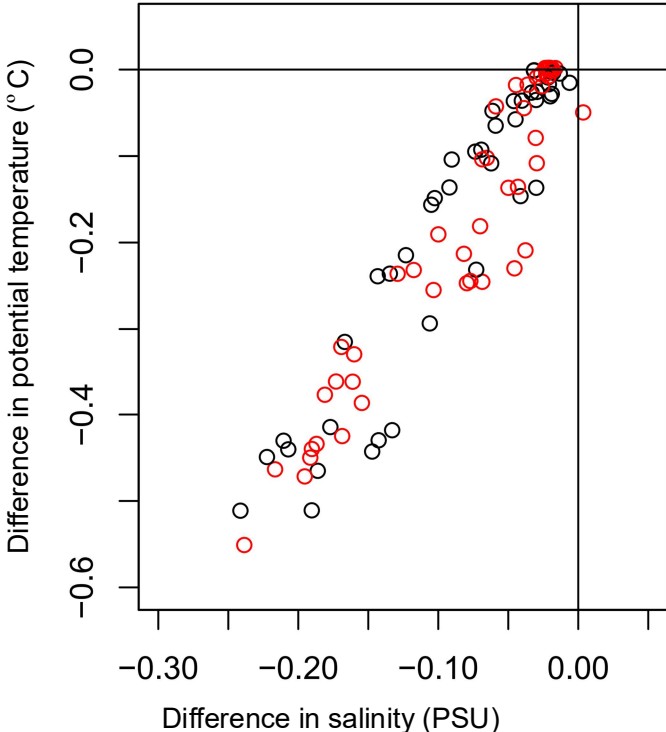

**Fig. 3. Salinity difference-temperature difference distribution of water in the RIS cavity (EI minus EN). The horizontal axis is for salinity difference and the vertical axis is for temperature difference. The inflow anomaly and outflow anomaly are marked with red and black, respectively. The units for salinity and temperature are PSU and ℃, respectively.**

## 3.2 Influence of BMRIS on the Southern Ocean

BMRIS contributes to salinity changes in the Southern Ocean (ocean south of 35º S; separation of the global ocean in 1 x 1 longitude-latitude grids is shown in Fig. S1). The area−averaged salinity decreases in water deeper than 1500 m in the Southern Pacific Ocean and the Southern Indian Ocean(Fig. 4). In the Southern Atlantic Ocean, the salinity increases in
10 water deeper than 4000 m. From the surface to 1500 m, the curves of salinity difference have similar shapes, whereas the curves of temperature difference do not. In the middle layer of the water body (approximately 1500–3000 m), the water in EI becomes colder and fresher due to the addition of cold and fresh water. In the deep ocean (deeper than 3100 m), the water in EI predominantly becomes warmer. In the shallow ocean (shallower than approximately 550 m), the temperature biases are more varied (see Fig. 4a-4c). At the sea bottom, water in most of the area south of 45° S becomes fresher (Fig. 5), which is
15 consistent with the results from Fig. 4; large differences mostly appear in the cavity of RIS (Fig. 5). The BMRIS has the

biggest influence on bottom water in the Southern Pacific Ocean, especially the Ross Sea and its adjacent western (looking from the north) deep ocean. The signal of the BMRIS effect is weak in the Southern Atlantic Ocean compared to those in the Southern Pacific Ocean and the Southern Indian Ocean. This result agrees with the results of thermohaline circulation, in which the deep current moves southward in the Atlantic Ocean.

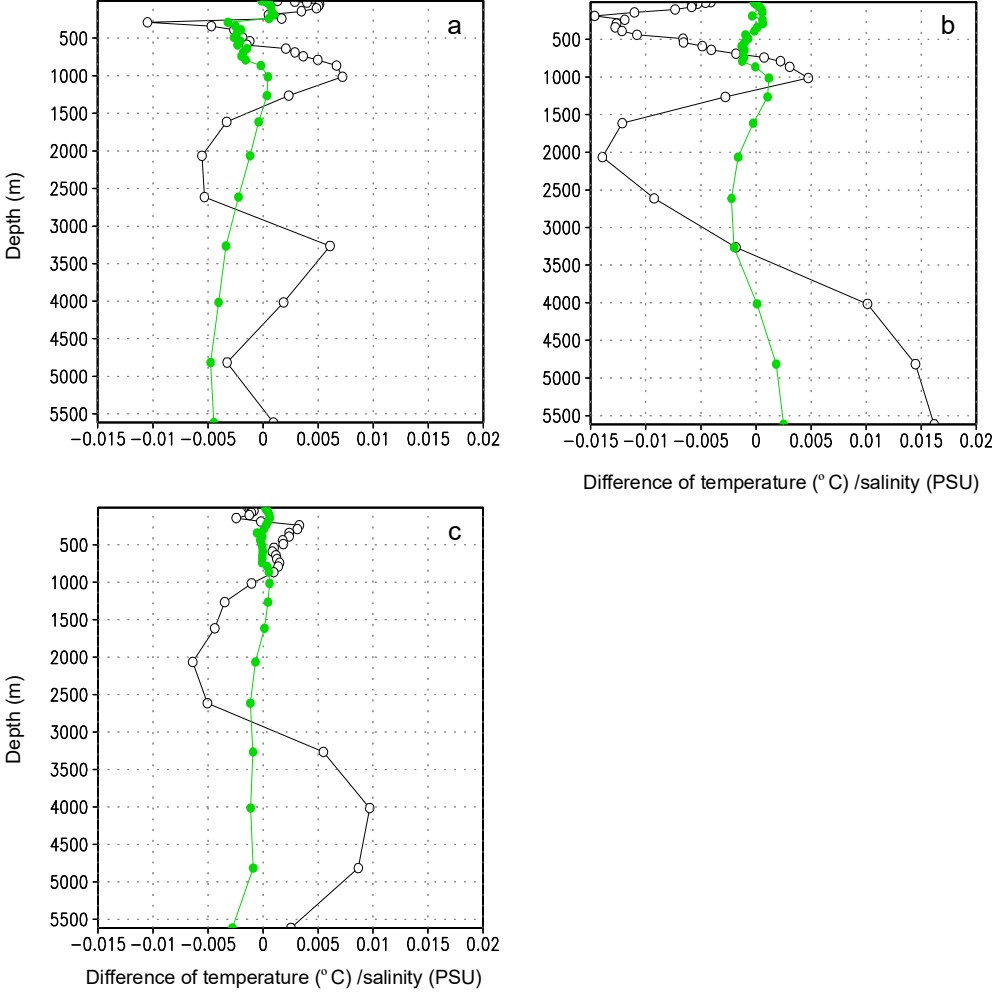

**Fig. 4. Area-averaged differences of salinity (solid circle) and potential temperature (open circle) (EI minus EN). The horizontal axis represents the difference and the vertical axis represents ocean depth. (a) Southern Pacific Ocean. (b) Southern Atlantic Ocean. (c) Southern Indian Ocean. The units for salinity and temperature are PSU and °C, respectively.**

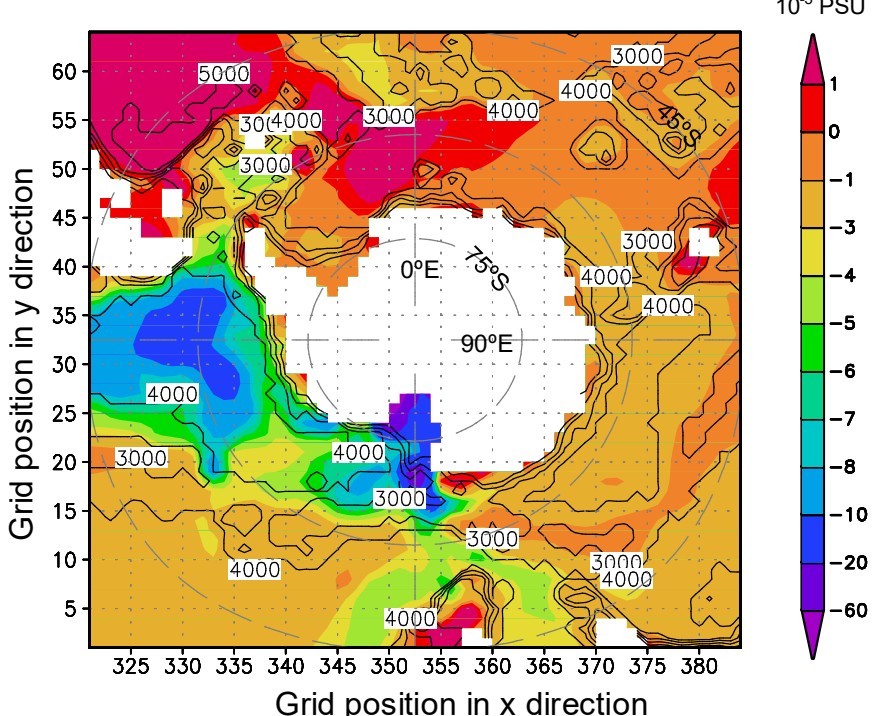

**Fig. 5. Annual mean salinity differences (EI minus EN, shaded) at the sea bottom. The contour lines represent the water thickness with intervals of 1000 m. The unit for salinity is PSU. It should be noted that the shaded contour intervals are uneven.**

10      Using the International Thermodynamic Equation of Seawater-2010 (TEOS-10), the total differential in density $\rho$ can be expressed as

$$d\rho = \frac{\partial \rho}{\partial \Theta}\Big|_{S_A,p}\, d\Theta + \frac{\partial \rho}{\partial S_A}\Big|_{\Theta,p}\, dS_A = \rho(-\alpha d\Theta + \beta dS_A),$$

where $\alpha = -\dfrac{1}{\rho}\dfrac{\partial\rho}{\partial\Theta}\big|_{S_A,p}$, $\beta = \dfrac{1}{\rho}\dfrac{\partial\rho}{\partial S_A}\big|_{\Theta,p}$; $S_A$ is absolute salinity, $\Theta$ is conservative temperature, $p$ is pressure, $\alpha$ is the coefficient of thermal expansion, and $\beta$ is the coefficient of haline contraction. Using the Gibbs Seawater Oceanographic Toolbox in Fortran (https://github.com/TEOS-10/GSW-Fortran), such variables as $S_A$, $\Theta$, $\alpha$, $\beta$, and $\rho$ can be easily computed. The $S_A$ difference-$\Theta$ difference distribution of water in the RIS cavity (EI minus EN) is similar to that in Fig. 3. In polar oceans, $\beta$ can be ten times larger than $\alpha$ (see Fig. S2). This implies that the change of density is dominated by that of salinity if temperature variation and salinity variation are in the same order. The added fresh water reduces the salinity in water near the RIS. This reduced salinity gives rise to a reduction in sea water density (Fig. S3).

BMRIS therefore adds a freshwater flux of 3.15 mSv (1 mSv=$10^3$ m$^3$ s$^{-1}$) to the upper ocean. This surplus of fresh water decreases the ocean water density and generates anticlockwise circulation anomalies in the Ross Sea. In addition, due to the topography effect near the location (65° S, 170° E), a clockwise circulation anomaly is induced and superimposed on the Antarctic circumpolar current (ACC) in EI (Fig. 6). The two circulation anomalies work together to produce a warm advection anomaly near the location (67° S, 180° E). Associated with this warm advection anomaly is a warm sea surface temperature (SST) anomaly (Fig. 6a). The cold water from BMRIS is advected clockwise by ACC, which contributes to a cold SST anomaly, an anomalous surplus of sea ice concentration (SIC), and a sea ice thickness (SIT) surplus over a broad area (Figs. 6 and 7). In the austral winter, the sea ice extent increases due to decreased SST. The anticlockwise circulation anomaly associated with the low water density anomaly in the Ross Sea and the circulation anomaly in the north forms a convergence anomaly at the margins of the ice cover, approximately along the latitude circle of 62° S. This leads to warm water accumulation and SST increases in EI (Fig. 6b), as well as SIC and SIT decreases (Fig. 7b). These observed SIT differences contrast to those of Hellmer (2004), in which SIT in the Ross Sea increases and shows no significant difference in ocean areas downstream the Ross Sea. His study reports the results of the 20$^{th}$ model year from a regional coupled ice-ocean model and the RIS cavity geometry is not included in the model bathymetry for the no sub-ice freshwater input experiment. Thus, differences between these studies may largely be due to the different treatments of RIS cavity geometry in the no sub-ice melting experiments. In this study, the difference of quasi-equilibrium states is discussed and the magnitude of the SIT difference is far smaller than that reported in Hellmer (2004).

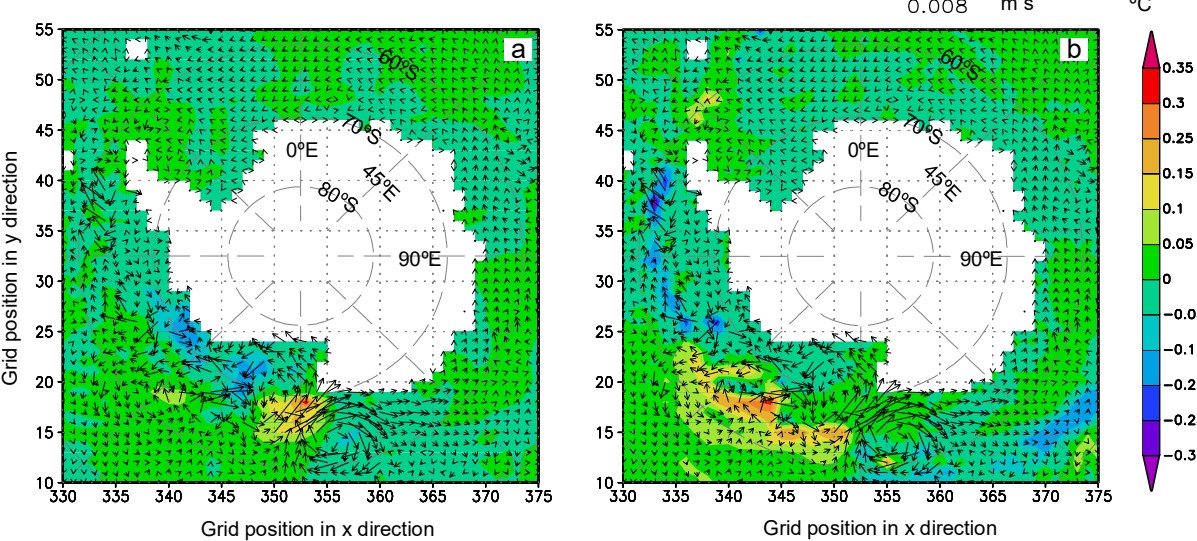

**Fig. 6. Differences of sea surface temperature (shaded contours) and current (arrows) (EI minus EN). (a) March. (b) September. The units for temperature and current are °C and m s⁻¹, respectively.**

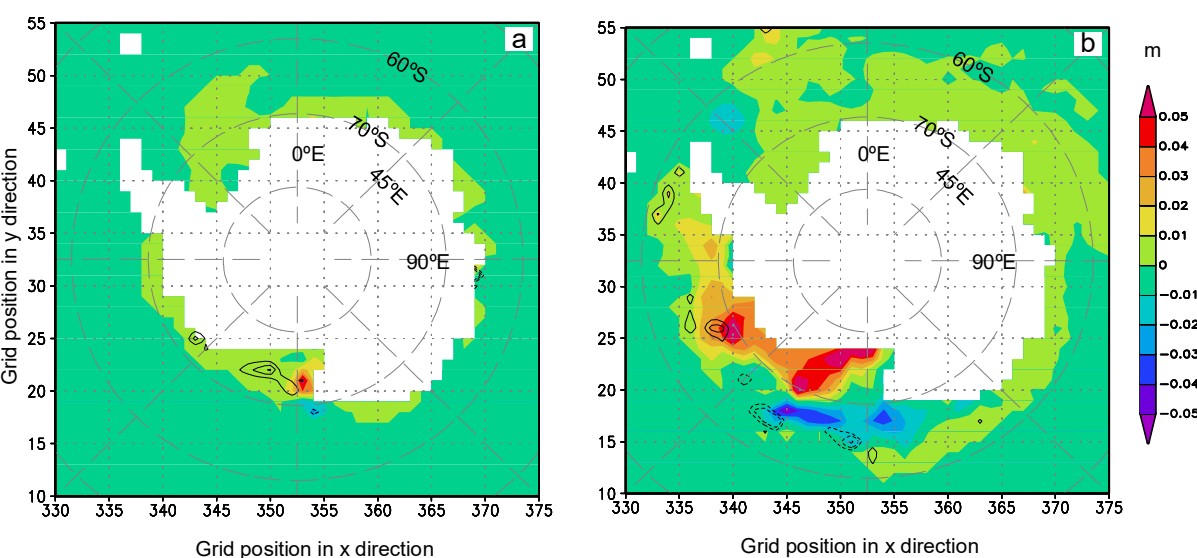

**Fig. 7. Differences of sea ice concentration (SIC) and sea ice thickness (SIT) (EI minus EN). (a) March. (b) September. The differences of SIT are shaded. The contours in black represent the differences of SIC, in which contour intervals are 0.02 and 0.05 for (a) and (b), respectively, and lines of 0 are not plotted. The units for SIC and SIT are 100% and m, respectively.**

Similar to the pattern of surface currents (Fig. 6), the ACC is weakened in the depth-averaged ocean currents in regions other than that north of the Ross Sea when the BMRIS effect is considered (Fig. 8). There are also two circulation anomalies near the Ross Sea. One is anti-clockwise and the other is clockwise. This circulation pattern is maintained until approximately 2000 m depth (Fig. S4), implying a combined influence of salinity difference from BMRIS and the characteristics of local bathymetry. As the density variation is dynamically linked with the flow, the ultimate pattern of ACC differences is the result of the mutual adjustment between the velocity and density fields.

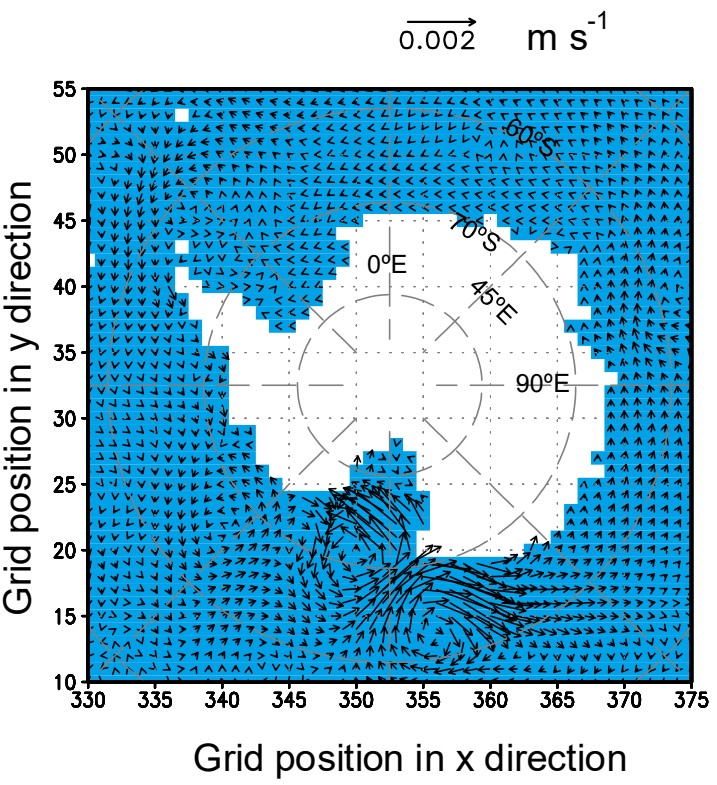

**Fig. 8. Differences of annual mean depth−averaged ocean currents (EI minus EN). The unit of velocity is m s⁻¹.**

The meridional overturning circulation (MOC), which is a system of surface and deep currents encompassing all ocean basins, is usually depicted by the meridional transport stream function. When the BMRIS effect is introduced, the strength of the Antarctic Subpolar Cell and Lower Cell weakens (Fig. 9a) (here, the cell names follow the convention of Farneti et al. (2015)). This is because the enrichment of fresh water from BMRIS decreases the water density and dampens the sink of surface dense water, thus significantly weakening the downward branch of MOC over the Antarctic continental slope. As a consequence, the formation and spreading of the AABW will also be influenced.

Because the meridional transport stream function in depth-latitude space cannot reflect real diapycnal transport in the ocean (the path of overturning circulation may parallel the contour of potential density in some places), it is recommended that it be evaluated in density-latitude space (Ballarotta, et al., 2013). When zonal integration is performed along potential isopycnals, the positions and strength of cells in the meridional-isopycnal frame cannot always be traced back to their counterparts in depth-latitude space. As seen in Fig. 9b (the calculation of potential density follows the algorithm of Jackett et al. (2006) and the reference pressure used here is 2000 dbar), there are more isolated cells that have no counterparts in Fig. 9a. The strength and position of the Subpolar Cell, Upper Cell, and Lower Cell in this model much more closely resemble those in ACCESS and GFDL-MOM given in Farneti et al. (2015). The strength and position of simulated cells in Farneti et al. (2015) are variable; the biggest discrepancy among the models exists in the strength of the anti-clockwise Lower Cell, which ranges from 20 Sv to zero. The simulated strength of the Lower Cell from EI is approximately 15 Sv. In density-latitude space, the Subpolar Cell and Lower Cell in the Southern Ocean also weakens (Fig. 9b).

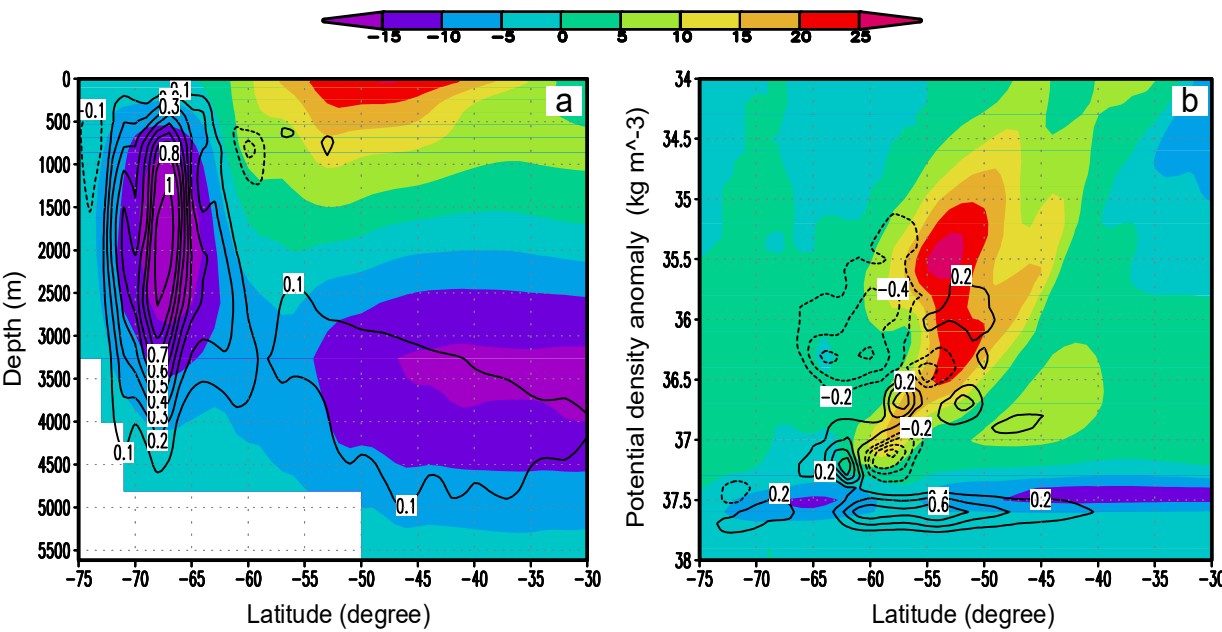

**Fig. 9. Meridional transport stream function of EI (shaded contours) and its difference from EN (EI-EN, contours): (a) in depth-latitude space and (b) in density-latitude space. The contour intervals for the meridional transport stream function difference in (a) and (b) are 0.1 Sv and 0.2 Sv, respectively, and the 0 Sv line is not plotted.**

The simulated heat transport in EI is similar to that derived from the NCEP reanalysis dataset given by Trenberth and Caron (2001) with the biggest difference located in latitudes around 55° S, where the simulated southward heat transport is approximately 0.2 PW weaker. The BMRIS contributes to the ocean heat transport anomaly by changing MOC. Considering

the global ocean as one water body, the BMRIS contributes to reduced southward heat transport in most latitudes of the Southern Ocean; the maximum reduction occurs at approximately 70° S (Fig. 10). Compared to the magnitude of the full heat transport, the maximum reduction of southward heat transport occurs at 71° S with an approximate value of 6%, whereas the relative reduction is less than 1% at most other latitudes.

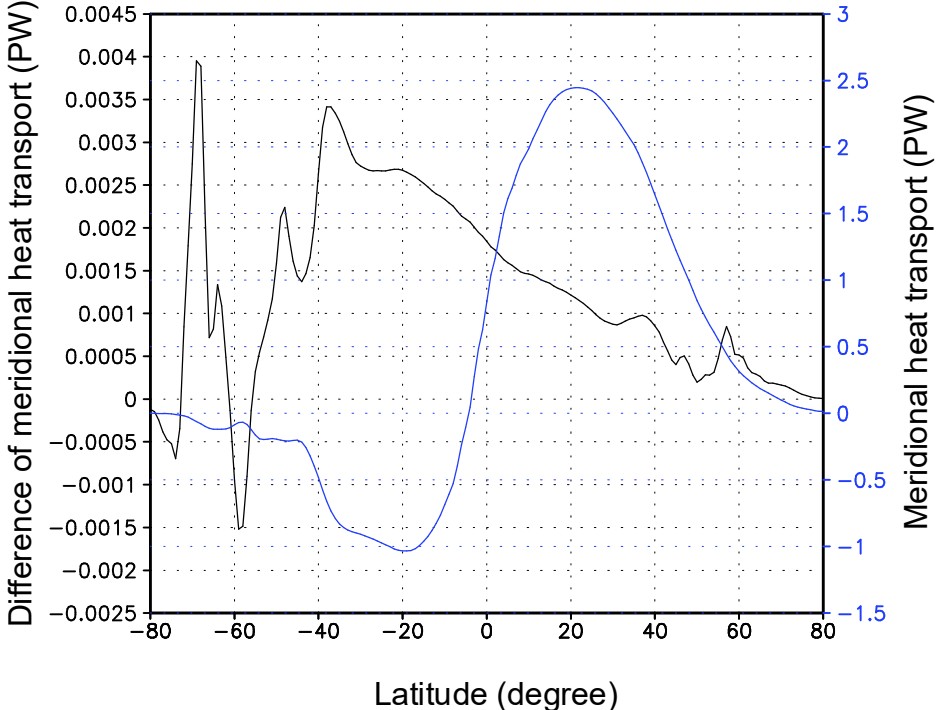

**Fig. 10. Meridional heat transport for the global ocean from EI (blue line on the right vertical axis) and its deviation from EN (EI minus EN, black line on the left vertical axis). The units for the vertical and horizontal axes are PW and degrees, respectively.**

**4 Conclusion and discussion**

10    Through numerical modeling, we studied the influences of BMRIS on the Southern Ocean in quasi-equilibrium. The aim of the study was to show that, through steady basal melting, the BMRIS leads to some interesting long-term phenomena. In quasi-equilibrium, the freshwater flux from BMRIS is 3.15 mSv, which is associated with a basal melt rate of 0.25 m a$^{-1}$. This freshwater decreases the salinity and density in the Antarctic Ocean. The decreased density from BMRIS together with the influence of ocean bathymetry generates local circulation differences in the Ross Sea and adjacent regions. The cold

anomaly from BMRIS is advected clockwise by ACC and then increases sea ice thickness and sea ice concentration in the affected region. In quasi-equilibrium, the strength of ACC in most areas except the northern part of the Ross Sea is reduced. The density anomaly from BMRIS stabilizes the water near Antarctica and weakens the sub-polar cell as well as the lower cell in the global MOC, which is accompanied by reduced southward meridional heat transport in most latitudes of the Southern Ocean.

According to a simulation study by Beckmann and Goosse (2003), in which ice shelf basal melting was parameterized as a function of oceanic temperature on the shelf/slope area of the adjacent ocean as well as an effective area of interaction, the basal melt rate of RIS differs substantially with different atmospheric forcing fields. Atmospheric forcing resolution affects the delivery of ocean heat to Antarctic floating ice shelves; higher-resolution winds can lead to more heat being delivered to the ice shelf cavities from the adjacent ocean and an increased efficiency of heat transfer between water and ice (Dinniman et al., 2015). Thus, simulations with other atmospheric forcing fields may be useful to ascertain the effects of BMRIS on the Antarctic Ocean.

In this study, ISOMIP thermodynamics were used, which neglect the velocity dependence of heat- and salt-transfer coefficients. In velocity-independent melt rate parameterizations, the impact of currents or tides on the distribution of sub-ice-shelf melting is indirect, and therefore limited (Dansereau et al.,2014). If the velocity dependence of transfer coefficients is considered, as in the most recent modeling studies using fine grids (Dansereau et al.,2014; Asay-Davis et al.,2017), differences in melt rate patterns may be observed. These differences may be more significant in higher resolution modeling because of the improved resolution of high boundary layer currents.

Ice shelves range in size from 500 000 km² (RIS) to approximately 100 km² (Ferrigno ice shelf). Current global ocean model configurations cannot explicitly resolve all the ice shelf cavities, especially in large-scale simulations. As illustrated by some studies (for example, Rignot et al., 2013; Nakayama et al., 2014), small ice shelves can produce significantly more freshwater than RIS and impact the Antarctic climate both locally and regionally in significant ways. Not all ice shelves are in a stable state (some are thickening and some are thinning) (Rignot et al., 2013). To study the long-term influences of stable ice shelf basal melting on the Southern Ocean, the RIS was included with an appropriate model resolution for long integration in this study. However, model horizontal resolution is important not only for simulating the conditions underneath the ice shelf that lead to basal melting but also for the open-ocean conditions that deliver heat to ice shelf cavities and for identifying relevant water masses (Dinniman et al.,2016; Little and Urban, 2016). Increasing the model resolution dramatically improves the representation of Circumpolar Deep Water on the Amundsen Sea continental shelf (Nakayama et al., 2014; Dinniman et al., 2015). More research using a finer resolution should be conducted to reduce the uncertainty in the simulation of the BMRIS effect on the Southern Ocean. Moreover, the effects of other ice shelves, such as the Filcher-Ronne ice shelf, should also be evaluated.

**Acknowledgments** I would like to thank X. Asay-Davis and R. Walker for their thorough review and helpful comments and suggestions. The experiments in the work were designed and performed during visits to C. M. Bitz at the University of Washington in Seattle. I would also like to thank X. Zhang for his helpful questions on the manuscript. This work is partly sponsored by the China Special Fund for Research in the Public Interest (Grant No. GYHY201506011) and the Natural Science Foundation of China (Grant No. 41276190).

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

**Table 1. An incomplete list of ice shelf-ocean coupled modeling studies**

| Publication | Ocean model | Ice shelf implementation | Domain and time periods covered |
|---|---|---|---|
| Beckmann and Goosse (2003) | Coupled Large-scale Ice Ocean (CLIO) | parameterization from an ice shelf-ocean interaction model | global, 100 years |
| Grosfeld and Sandhager, (2004) | Rigid-lid, hydrostatic primitive equation model, formulated in spherical coordinates | dynamic | 900 km x 700 km in the horizontal, 300 years |
| Hellmer (2004) | Bremerhaven Regional Ice Ocean Simulations (BRIOS) | thermodynamics with fixed cavity | circumpolar, 20 years |
| Walker and Holland (2007) | A two-dimensional model in the yz-plane | simplified dynamic | 600 km x 1100 m, 600 years |
| Losch (2008) | MIT general circulation model (MITgcm) | thermodynamics with fixed cavity | in ISOMIP (Ice Shelf–Ocean Model Intercomparison Project) experiment: from 0º E to 15º E and 80º S to 70º S, 10 000 days<br><br>in (nearly) global ocean model (excluding the Arctic Ocean) experiment: 80º N southward, 100 years |
| Timmermann et al. (2012) | Finite Element Sea-ice Ocean Model (FESOM) | thermodynamics with fixed cavity | global, 53 years |
| Galton-Fenzi et al. (2012) | Regional Ocean Modeling System (ROMS) | thermodynamics with fixed cavity | regional, 20 years |
| Kusahara and Hasumi (2013) | Sea ice-ocean coupled model (COCO) | thermodynamics with fixed cavity | circumpolar, 25 years for CTRL run and 38 additional years for ERA-INT case |
| Mathiot et al. (2017) | Nucleus for European Modeling of the Ocean (NEMO) | thermodynamics with fixed cavity | in academic case: from 0º E to 15º E and 80º S to 70º S, 10 000 days<br><br>in real ocean application: circumpolar, 10 years |

**Table 2. Major parameters for the *shelfice* package used in EI**

| Parameter | Value | Parameter | Value |
|---|---|---|---|
| Heat transfer coefficient that determines heat flux into ice shelf | $10^{-4}$ m s$^{-1}$ | Salinity transfer coefficient that determines the salt flux into the ice shelf | $5.05\ 10^{-7}$ m s$^{-1}$ |
| If a simple ISOMIP (Ice Shelf-Ocean Model Intercomparison Project) thermodynamics is used | yes | If conservative ice-ocean interface boundary condition following Jenkins et al. (2001) is used | no |
| If average over boundary layer width is used | yes | If slip condition for ice shelf is used | yes |

**Table 3. Basal melt rates averaged over the entire RIS derived in this study and other studies**

| Basal melt rates (m a$^{-1}$) | Source | Brief description |
| --- | --- | --- |
| 0.12 ± 0.03 | Shabtaie and Bentley (1987) | Calculated from the measured ice flux into the Ross Ice Shelf and previous measurements |
| 0.18–0.27 | Hellmer and Jacobs (1995) | Calculated from a two-dimensional (y-z plane) channel flow model forced by density differences between the open boundaries and the interior cavity |
| 0.25 | Assmann et al. (2003) | Calculated from a circumpolar numerical model |
| 0.082 | Holland et al. (2003) | Calculated from a regional numerical model (MICOM) |
| 0.13–0.15 | Dinniman et al. (2007) | Calculated from a regional numerical model (ROMS) |
| 0.15 | Dinniman et al. (2011) | Calculated from the ROMS model |
| 0.6 | Timmermann et al. (2012) | Calculated from a global finite element ocean model (FESOM) |
| 0.0± 0.1 for Ross West<br><br>0.3 ± 0.1 for Ross East | Rignot et al. (2013) | Calculated from radar measurements and output products from the Regional Atmospheric and Climate Model RACMO2 |
| 0.14 ± 0.05 | Depoorter et al. (2013) | Calculated from radar measurements and a regional climate model (for firn air content and compaction) |
| 0.25 (without tidal forcing)<br><br>0.32 (with tidal forcing) | Arzeno et al. (2014) | Calculated from the ROMS model |
| 0.11 ± 0.14 (converted from the basal melt budget of RIS dM/dt in Table 3 with an ice density of 918 | Moholdt et al. (2014) | Derived from Lagrangian analysis of ICESat (NASA's Ice, Cloud and Land |

| | | |
|---|---|---|
| kg m$^{-3}$) | | Elevation Satellite) altimetry |
| 0.24 (converted from basal melt in Gt a$^{-1}$ for the last year of simulation in R_MLT in Table 3 with an RIS area of 500 000 km$^2$ and an ice density of 918 kg m$^{-3}$) | Mathiot et al. (2017) | Calculated from a regional numerical model (NEMO) |
| 0.25 | This study | Calculated from quasi-equilibrium state of a global numerical model (MITgcm) |