# Peer review of "Modeling the effect of Ross Ice Shelf melting on the Southern Ocean in quasi-equilibrium"

_The Cryosphere, 2017_

## Referee Comment (RC1) · R. Walker (Referee) · 6 Jan 2018

Inclusion of ice shelves in global circulation models is a significant issue for the accuracy of climate projections. This study considers the impact of basal melting under the Ross Ice Shelf on the Southern Ocean by contrasting global ocean model experiments with and without melting in the sub-Ross cavity. The choice of a no-melt scenario that includes sub-ice-shelf bathymetry seems a little odd to me, as most ocean modeling that I'm aware of either includes ice shelves plus melting or excludes ice shelves from the domain. It should still be possible to get value from this experimental setup. However, I would have liked this manuscript to spend much more time on detailed discussion of the different experiments, particularly the relations between water properties and dynamics.

[Figure]

General comment on figures) All units should be in axis labels, not only in the captions. Also, axes should be labeled with variable names. Figures 1, 2, 6, 7, 9, 10 should have a larger font size to be readable.

Page 2: Line 9) "The equivalent freshwater flux..." This is unclear. Do you mean that the freshwater flux is equivalent to a particular melt rate over the ice shelves?

Figure 1b) On my printout, this looks like green, not yellow.

Section 3.1) Is the first paragraph about both experiments or only EI?

5:5) "The difference in the feature ..." This calls for more explanation.

5:10) When listing the earlier results, it would be good to provide the actual numbers for comparison.

5:14) "The difference in seasonality ..." Also could use more explanation.

Figure 3) Write out the full names of the variables in the axis labels.

7:6) What latitudes are you considering to be the Southern Ocean?

7:15) This could use a description of the complex mechanisms.

7:17) What happens in the Southern Atlantic?

7:19) Why aren't you showing the figure? I don't think there's a limit on number of figures here.

Figure 4) This would be easier to read with the y-axis flipped so the surface is at the top of the graph.

Figure 5) The color scale here doesn't show detail over most of the domain because of a few outliers under the Ross. Probably would be better to plot Ross separately or just discuss the values there in the text.

9:7) Describe the specific bathymetry feature.

9:16) It would be better to compare your output with Hellmer's for the case of ice-shelf melt being included. The difference you're describing here is more or less a matter of how you define the no-melt experiment setup.

Figures 6 and 7) The color scales for the subplots should be equal for (a) and (b). Also, the arrows in Figure 6 are very small and hard to read.

11:2) Again, why not show the figure?

Figure 8) You may want to zoom in to show the gyres better.

11:15) Could use a reference for the recommendation.

Figure 9) The contours of the difference overlying the EI shaded contours are hard to follow, at least for me. The difference could use its own subplot.

12:18) It would be useful to compare the heat transport anomalies to the magnitude of the full heat transport.

Figure 10) Cut "stream function" in caption.

13:7) For consistency with the rest of the paper, this should be Southern Ocean.

---

## Referee Comment (RC2) · X. Asay-Davis (Referee) · 29 Jan 2018

First, I deeply apologize for taking significantly longer than requested to review this manuscript. I hope my thorough review helps to mitigate the inconvenience of waiting so long for my review.

General Comments:

This paper describes two global ocean-sea ice experiments run to quasi-equilibrium over 500 years, one with basal melting below the Ross Ice Shelf (RIS) and one without basal melting (but still with an ocean cavity below RIS). Most of the presented results examine differences between ocean and sea ice properties between the two experiments (results are typically averaged over the last 100 years of each experi-

ment). Non-negligible differences in the horizontal and vertical distributions of temperature and salinity are found between the two experiments, leading to appreciable differences in both the meridional overturning circulation and meridional heat transport in the ocean. Differences in the surface and barotropic flow are also demonstrated, along with related changes in sea surface temperature, sea ice thickness and sea ice concentration.

The purpose of the study seems to be to show that freshwater fluxes from ice shelves have a significant impact on the Southern Ocean compared to a simulation without any freshwater fluxes. It seems unclear (and is not discussed in the manuscript) what the relevance of this study is to other modeling work or what this study might tell us about the melt-induced dynamics in the real world.

It has been known for some time in the Earth System Modeling community that some form or freshwater input into the deep ocean is required for adequate representation of deep ocean properties and of the meridional overturning circulation. Therefore all global Earth System Model (ESM) include some mechanism for freshwater input (typically surface "runoff" around Antarctica and Greenland), together with a mechanism for inducing overturning in polar regions (typically salinity restoring at the surface). No models I am aware of without sub-ice-shelf melting would leave out these mechanisms. Therefore, if the aim of this study is to show that current ESMs should be including the effects of ice-shelf melting in order to avoid inaccuracies in Southern Ocean properties, the "control" experiment (EN in the manuscript) should probably have been closer to a configuration used in ESMs: "runoff" at the surface to at least partially account for freshwater fluxes and no ice-shelf cavities.

If the purpose of the study is to show what features of the climate system are affected by the presence or absence of melting below RIS, there is another significant pitfall in this work. Very little effort is made to validate either the EI (with melting) or the EN (without melting) experiment against observations or previous modeling (except for the basal melt rate below RIS). This strikes me as highly problematic because the differences between the simulations is unlikely to tell us something about the real world if the base state (either EI or EN) that is being perturbed can be shown to be representative of the real world. Given the \*very\* coarse horizontal resolution (150 km) and rather coarse vertical resolution (not stated but seemingly around 50 m), it seems unlikely that the model will be able to capture the complex chain of processes by which water masses are transformed on the Ross continental shelf, within the Ross cavity, and off the continental slope where they mix into the deep ocean. These processes have been shown to require horizontal resolutions at least 30x higher than this simulation (see specific comments), allowing interactions between small-scale topographic features and narrow oceanic currents. Without these transformations being captured adequately or the model state having been validated against a broader set of observations, conclusions in this work about how basal melting affects the Southern Ocean are likely to only apply to this particular model configuration, and not to be representative of the real world.

The manuscript presents much of the results results with little deeper analysis, discussion or explanation (the exception is a more careful analysis changes in sea surface temperature and sea ice properties resulting from flow anomalies near RIS). Except for the dynamics at the ocean surface, little attempt is made to explain how water masses are transformed to reach various ocean depths. Basal melting is found to \*decrease\* the global overturning circulation, seemingly due to increased stabilization of the water column, in contradiction to know physical processes of Antarctic Bottom Water formation (known to occur in the Ross Sea region) that are thought to be an important driver of global ocean circulation. No discussion is included of potential shortcomings of the model at capturing or resolving ocean processes that would be relevant to these transformations.

I can only recommend this manuscript for publication after major revisions to address these shortcomings.

This paper would benefit from significant editing by a native English speaker. I have

attempted to point out typos and grammatical errors where I have seen them (I include about 3 pages of such corrections). Additionally, the figures all need significant formatting work before they are ready for publication, including labeling axes and increasing font sizes to make the labels more readable.

Specific Comments:

p. 1 l. 6: In the field, BMR is typically used as an abbreviation of "basal melt rate". The incorporation of the Ross Ice Shelf into this abbreviation is confusing. I would suggest replacing "basal melting of Ross Ice Shelf (BMR)" with "basal melting below the Ross Ice Shelf (Ross BM)" and elsewhere replace "BMR" with "Ross BM" to avoid confusion. If you can come up with an alternative shorthand that will not be confused with "basal melt rate", that would be fine, too.

p. 1 l. 12: I would suggest replacing "substantially" and "not so significant" with something more quantitative if possible.

p. 1 l. 14: "local circulation anomalies": In general, the abstract seems to treat the case of no basal melting as the control case and the case with basal melting as the modified experiment. I can understand this choice, since ocean models typically do not include ice-shelf cavities, though it seems strange from a physical standpoint to treat the less physical experiment as the control case. Here, the use of the word "anomalies" seems particularly strange to me, since it seems to imply "something that deviates from what is standard, normal, or expected", whereas I would say the control case is the one more likely to deviate from the physical world. Perhaps another phrase such as "differences in local circulation) would be clearer

p. 1 l. 14: "with the help of ocean bathymetry": This phrase seems rather vague to me. Maybe a better wording would be something like "The decreased density due to the effect of Ross BM, together with interactions with ocean bathymetry, creates local differences in circulation in the . . ."

[Figure]

p. 1 l. 22-24: The audience for The Cryosphere is aware of what ice sheets, ice shelves, icebergs, etc. are so I don't think this level of introduction is necessary.

p. 1 l. 26: "beneath the currently stable Ross Ice Shelf": The phrase "currently stable" is both grammatically problematic and confusing, because it implies a past or future instability in RIS that is not addressed here, nor is there any widely accepted likelihood of RIS instability in the community. I would remove this phrase.

p. 1 l. 26: "can be larger than 2500% of the overall. . .": It is not clear that this fact or this reference is relevant to the rest of the paper, as you are not resolving melt channels in your simulations.

p. 2 l. 3: "Neglecting the sub-ice freshwater...for the Southern Ocean." While it is not stated here, the implication seems to be that common practice in ocean modeling of the Southern Ocean is to neglect sub-ice-shelf freshwater fluxes entirely, whereas this is not usually the case. Global (and I believe also regional Antarctic) ocean models without ice-shelf cavities still include an approximation of the total Antarctic freshwater input (melting + calving) but they almost universally do so by distributing the freshwater at the ocean surface and typically evenly around the continent. In my view, sub-ice-shelf freshwater fluxes aren't really "neglected" so much as they are estimated and distributed inaccurately. Here is one publication that discusses the differences in ocean model behavior depending on how freshwater fluxes are distributed: Mathiot, P., Jenkins, A., Harris, C., and Madec, G.: Explicit representation and parametrised impacts of under ice shelf seas in the zâĹŮ coordinate ocean model NEMO 3.6, Geosci. Model Dev., 10, 2849-2874, https://doi.org/10.5194/gmd-10-2849-2017, 2017.

p. 2 l. 3-4: "These are pronounced in the Weddell...broad continental shelves". It is not clear to me that the Weddell and Ross Seas are the regions of Antarctica that would be most affected by neglecting freshwater fluxes. The large size of RIS and Filchner-Ronne Ice Shelf (FRIS) together with their relatively cold ice-shelf cavities do make them particularly important for AABW formation but other regions of Antarctica

with warmer cavities have been shown to produce significant amounts of freshwater that impact Antarctic climate both locally and regionally in significant ways, see e.g.:

Nakayama, Y., R. Timmermann, C. B. Rodehacke, M. Schröder, and H. H. Hellmer (2014), Modeling the spreading of glacial meltwater from the Amundsen and Bellingshausen Seas, Geophys. Res. Lett., 41, 7942–7949, doi:10.1002/2014GL061600.

The Getz, Thwaites and Pine Island Ice Shelves, for example, each produce significantly more freshwater than RIS and nearly as much as FRIS, despite their significantly smaller areas:

Rignot, E., Jacobs, S., Mouginot, J., Scheuchl, B. Ice-shelf melting around Antarctica. Science. 2013 Jul 19;341(6143):266-70. doi: 10.1126/science.1235798.

p. 2 l. 8-9: It would be good to supply a more complete list of estimates of basal melting. Here are a few more important ones:

Moholdt, G., L. Padman, and H. A. Fricker (2014), Basal mass budget of Ross and Filchner-Ronne ice shelves, Antarctica, derived from Lagrangian analysis of ICESat altimetry, J. Geophys. Res. Earth Surf., 119, 2361–2380, doi:10.1002/2014JF003171.

M. Depoorter, J. Bamber, J. Griggs, J. Lenaerts, S. Ligtenberg, M. van den Broeke, G. Moholdt. Calving fluxes and basal melt rates of Antarctic ice-shelves. Nature, 502 (7469) (2013), pp. 89-92

p. 2 l. 10: Other sources (Rignot et al 2013, Depoorter et al. 2013) estimate a significantly larger mean melt rate on the order of 0.8-0.9 m/a. Beckmann and Goosse, 2003 is not really an appropriate citation for the 0.5 m/a number, they are merely citing the Jacobs et al. 1996 estimate, converted from mSv to m/a. Given the significant improvements in satellite observations since 1996, I do not feel that number is particularly trustworthy.

p. 2 l. 11: "occurs at the base of the ice shelf edge": This is sometimes true, particularly for warm ice-shelf cavities. But the freshwater plume in cold cavities typically reaches

neutral buoyancy at depths significantly below the ice-shelf edge:

Jacobs, S S, H H Helmer, C S M Doakea, A Jenkins, and R M Frolich. "Melting of Ice Shelves and the Mass Balance of Antarctica." Journal of Glaciology 38, no. 130 (1992): 375–87. doi:10.3198/1992JoG38-130-375-387.

For the purposes of the point you are making, it would be sufficient to say, "Since the injection of this freshwater occurs at depth rather than at the ocean surface..."

p. 2 l. 16-17: "can provide no direction information about sub-ice shelf circulation": This is not entirely true, as sub-ice-shelf observations include velocity measurements that can be used to infer at least some basic information about the sub-ice-shelf circulation. Temperature and salinity measurements can also be used to infer, through the fraction of Ice Shelf Water, the degree of interaction with the ice-shelf base, which also can provide information about the broad sub-ice-shelf circulation. I would suggest toning this down to say that it is difficult to infer the sub-ice-shelf circulation from borehole observations.

p. 2. l. 15-29: The citations in this paragraph seems out of date and incomplete. These reviews provide many citations that could help to fill in the gaps:

Dinniman, Michael, Xylar Asay-Davis, Benjamin Galton-Fenzi, Paul Holland, Adrian Jenkins, and Ralph Timmermann. "Modeling Ice Shelf/Ocean Interaction in Antarctica: A Review." Oceanography 29, no. 4 (December 2016): 144–53. https://doi.org/10.5670/oceanog.2016.106.

Asay-Davis, Xylar S., Nicolas C. Jourdain, and Yoshihiro Nakayama. "Developments in Simulating and Parameterizing Interactions Between the Southern Ocean and the Antarctic Ice Sheet." Current Climate Change Reports, October 24, 2017, 1–14. https://doi.org/10.1007/s40641-017-0071-0.

I would suggest a complete rewrite of the paragraph with a more complete list of the numerical methods, domains, time periods covered, etc.

[Figure]

In particular, there are several studies that have used the MITgcm with ice-shelf cavities in regional configurations to study Antarctica and the Southern Ocean. Since these use the same model as this study, it would seem like they might get particular emphasis here.

p. 2 l. 19: "dynamic": this could use further clarification. I think you mean dynamic ice-shelf geometry? How is this different from Walter and Holland (2007)?

p. 2. l. 21: "fixed cavity and thermodynamics": The cavity geometry is fixed but the thermodynamics is not – melt rates evolve with changing ocean conditions.

p. 2 l. 21: "parameterization": Again, more details on what this means would be helpful.

p. 2 l. 22-23: What would the other options be besides the list given? Global? Indeed, there are several studies with global models (Losch, 2008; Helmer et al. 2012; Timmermann et al. 2012, etc.)

p. 2 l. 23: "two-dimensional" needs more clarification – one horizontal dimension and one vertical.

p. 2. l. 28-29: "At present, this kind of research has rarely been reported." I think it is fair to say that this has not been done before.

p. 2 l. 30-p. 3 l. 6: Again, I think this paragraph is missing some important work. Many modeling efforts not mentioned here include the Ross Sea in larger regional or global models that are big enough to look at the effect of RIS on the Southern Ocean. Two examples are:

Timmermann, Ralph, and Hartmut H. Hellmer. "Southern Ocean Warming and Increased Ice Shelf Basal Melting in the Twenty-First and Twenty-Second Centuries Based on Coupled Ice-Ocean Finite-Element Modelling." Ocean Dynamics 63, no. 9–10 (October 2013): 1011–1026. https://doi.org/10.1007/s10236-013-0642-0.

Dinniman, Michael S., John M. Klinck, Eileen E. Hofmann, and Walker O. Smith. "Effects of Projected Changes in Wind, Atmospheric Temperature, and Freshwater Inflow on the Ross Sea." Journal of Climate, December 1, 2017. https://doi.org/10.1175/JCLI-D-17-0351.1.

You are correct that these models were not able to run for long enough times to look at quasi-equilibrium effects.

p. 3 l. 12: "will be an interesting topic": I don't think this belongs here, as it is a very subjective statement. I would remove this whole sentence.

p. 3 l. 17-19: Both the topography data and the forcing data are not the most up-to-date versions, see references below. Both Bedmap2 (Fretwell et al. 2013) and RTOPO2 (Schaffer et al. 2016) have updated topography, though I am not sure whether these changes affect RIS specifically. There is a CORE-NYF.v2 data set (http://data1.gfdl.noaa.gov/nomads/forms/core/COREv2/CNYF_v2.html), which is a climatology from the interannual forcing described in Large and Yeager (2009). It would be worth explaining why these earlier versions were used instead of the more up-to-date versions.

Fretwell, P, H D Pritchard, D G Vaughan, J L Bamber, N E Barrand, R Bell, C Bianchi, et al. "Bedmap2: Improved Ice Bed, Surface and Thickness Datasets for Antarctica." The Cryosphere 7, no. 1 (2013): 375–93. https://doi.org/10.5194/tc-7-375-2013.

Schaffer, Janin, Ralph Timmermann, Jan Erik Arndt, Steen Savstrup Kristensen, Christoph Mayer, Mathieu Morlighem, and Daniel Steinhage. "A Global, High-Resolution Data Set of Ice Sheet Topography, Cavity Geometry, and Ocean Bathymetry." Earth System Science Data 8, no. 2 (October 2016): 543–57. https://doi.org/10.5194/essd-8-543-2016.

Large, W. G., and S. G. Yeager. "The Global Climatology of an Interannually Varying Air–sea Flux Data Set." Climate Dynamics 33, no. 2–3 (August 2009): 341–64. https://doi.org/10.1007/s00382-008-0441-3.

p. 3 l. 17-19: How is "runoff" handled in each experiment (EI and EN)? I believe CORE specifies a runoff field that inputs freshwater into the Antarctic region equally around the continent and at the ocean surface at a level that is supposed to roughly match the surface accumulation over the continent (therefore accounting for the combined effect of runoff, sub-ice-shelf melting and calving, assuming AIS is in equilibrium). Was this runoff field included in your simulations?

p. 3 l. 24-26: I would suggest making this sentence a footnote.

p. 3. l. 27-28: Please explain the abbreviations "EI" and "EN".

p. 3 l. 29-30: More detail should be given about what the vertical resolution actually is. What is the resolution at the surface? At 1000 m depth? The coarsest resolution (at depth)? I suspect that, even with finer resolution in the upper 1000 m, 30 layers is inadequate to resolve the sub-ice-shelf plume in detail. Finer resolution would likely lead to a significantly different answer, see:

Losch, M. "Modeling Ice Shelf Cavities in a z Coordinate Ocean General Circulation Model." Journal of Geophysical Research 113, no. C8 (August 2008): 1–15. https://doi.org/10.1029/2007JC004368.

Schodlok, Michael P., Dimitris Menemenlis, Eric Rignot, and Michael Studinger. "Sensitivity of the Ice-Shelf/Ocean System to the Sub-Ice-Shelf Cavity Shape Measured by NASA IceBridge in Pine Island Glacier, West Antarctica." Annals of Glaciology 53, no. 60 (2012): 156–162. https://doi.org/10.3189/2012AoG60A073.

Schodlok, M. P., D. Menemenlis, and E. J. Rignot. "Ice Shelf Basal Melt Rates around Antarctica from Simulations and Observations." Journal of Geophysical Research: Oceans 121, no. 2 (February 2016): 1085–1109. https://doi.org/10.1002/2015JC011117.

p. 4 l. 2: "the horizontal resolution is about 150 km". This is one of my biggest concerns about this work. I realize that long time integrations are expensive but this coarse

resolution (coarser even than CMIP5 and CMIP6 models of the region) seems \*far\* too coarse to capture the relevant dynamics for the Antarctic region, most importantly the pathways for transporting freshwater from the RIS to the Southern Ocean. See the following paper for a discussion of the pathways and the resolution ($\sim$5 km) required to capture them:

Dinniman, Michael S., John M. Klinck, Eileen E. Hofmann, and Walker O. Smith. "Effects of Projected Changes in Wind, Atmospheric Temperature, and Freshwater Inflow on the Ross Sea." Journal of Climate, December 1, 2017. https://doi.org/10.1175/JCLI-D-17-0351.1.

See this paper for a discussion of the inadequacy of CMIP5 models at capturing Antarctic continental shelf processes:

Little, Christopher M., and Nathan M. Urban. "CMIP5 Temperature Biases and 21st Century Warming around the Antarctic Coast." Annals of Glaciology 57, no. 73 (September 2016): 69–78. https://doi.org/10.1017/aog.2016.25.

p. 4 Table 1: Please reformat values in scientific notation rather than "e" notation used in programming languages (e.g. $1.0 \times 10^{-4}$ if you are using LaTex). Here and elsewhere, "m/s" should be "m s^{-1}" and similarly "m/a" should be "m a^{-1}", etc.

p. 4 Table 1: Could you explain the choice to use ISOMIP thermodynamics? Neglecting the velocity dependence of the heat- and salt-transfer coefficients has been shown to reduce the accuracy of melt fields, see discussions in:

Dansereau V, Heimbach P, Losch M. Simulation of subice shelf melt rates in a general circulation model: velocity-dependent transfer and the role of friction. Journal of Geophysical Research: Oceans. 2014;119(3):1765–90. https://doi.org/10.1002/2013JC008846.

Asay-Davis, Xylar S., Nicolas C. Jourdain, and Yoshihiro Nakayama. "Developments in Simulating and Parameterizing Interactions Between the Southern Ocean and the

Antarctic Ice Sheet." Current Climate Change Reports, October 24, 2017, 1–14.
https://doi.org/10.1007/s40641-017-0071-0.

p. 4 Table 1: Could you explain why the Jenkins et al. (2001) form was not used? They
show that this can lead to a drift away from the expected linear relationship between T
and S over long timescales, which seems problematic given that this study is focused
precisely on long timescales.

p. 4 l. 11-12: Could you please explain the choice to remove the ice shelf cavity in
the 4 grid boxes rather than thicken the cavity? What criterion was used to decide
whether the cavity is too thin and should be set to zero? How does the cavity thickness
in the model compare with that of the original RTOPO-1 data set, averaged over each
grid cell? Was the cavity thickness increased in some cells to match some required
threshold (e.g. the column is more than x cells thick)? If so, was the ice draft moved
up or was the bathymetry moved down, or both? What is the area of the modeled
cavity compare to the area in RTOPO-1 and what would you expect the effect of this
difference to be (I would expect the modeled cavity is much smaller and that this would
lead to a reduced freshwater flux but a similar melt rate to observations). In summary,
more explanation of the method is needed.

p. 4 l. 12: I believe "depth" actually refers to "water-column thickness". Is that correct?
If so, please make this substitution.

p. 4 Fig. 1: "indicate grids where cavities are resolved". Since 4 grid boxes have water-
column thicknesses of zero, I would argue those grid boxes don't resolve the cavity and
should probably be removed from the figure or shaded differently.

p. 4 Fig. 1: I am deeply concerned that RIS, the main focus of the study, is captured
by only 15 grid cells and with seemingly 50 m vertical resolution and seemingly without
partial bottom cells (though neither of these are discussed in the text). The introduction
suggests that it is important to capture the sub-ice-shelf flow in models because it
cannot be observed directly, but such coarse resolution seems entirely inadequate to

do that job.

p. 5 l. 9-13: It would be helpful to have a figure, panel of a figure or table to compare these various melt rates. It would be useful to be more quantitative than "larger" and "smaller". It would also be important to separate results derived from modeling from those derived from satellite measurements. It is encouraging that the melt rate lies within the range of observational and previous model estimates. What about freshwater fluxes (given that the area of RIS in the model is probably significantly different from observations)? How do these compare with other studies?

p. 5 l. 13: What is meant by "net melt rate"?

p. 5 l. 15: What is meant by "model system evolution stage"? Does this refer to the numerical methods used to discretize the equations of motion?

p. 6 l. 7-8: "salinity bias and temperature bias": I don't think "bias" is the correct word here, as this would assume that the control case (without melting) are the observations, which they most certainly are not. I would also suggest avoiding the word anomaly unless you make clearer why you have chosen the EN experiment to be the "control" (implying you expect it to be the "normal" case in some sense). I think the most correct term, free from value judgments, would simply be "difference". So the sentence should probably read something like "The relationship between salinity and temperature differences in RIS cavity water between the two experiments..."

p. 6 l. 7-11: This linear relationship between T and S resulting from melting is well known and is called the Gade line:

Gade, HG. "Melting of Ice in Sea Water: A Primitive Model with Application to the Antarctic Ice Shelf and Icebergs." Journal of Physical Oceanography 9 (1979): 189–198.

It would be important to show if your line has the expected slope for a Gade line. Otherwise, it could indicate something is amiss with the sub-ice-shelf boundary conditions.

[Figure]

p. 6 l. 10: "ppt" should probably be "PSU", which is slightly different. I do not believe MITgcm uses ppt to measure salinity.

p. 6 l. 11: "there seems to be no significant influence on the inflow and outflow in the cavity": The only way I can make sense of this phrase is if "influence on" is changed to "difference between". Melting clearly has an influence on both the inflow and the outflow so it is clearly not correct to say there is no influence. I would suggest that this finding deserves more discussion. The only way to make sense of this is that, in quasi-equilibrium, a significant amount of outflowing freshwater recirculates into the cavity. This is a somewhat surprising finding and I think possibly a significant difference between these simulations and those at higher resolution (e.g. Nakayama et al. 2014, Dinniman et al. 2017):

Nakayama, Y., R. Timmermann, C. B. Rodehacke, M. Schröder, and H. H. Hellmer. "Modeling the Spreading of Glacial Meltwater from the Amundsen and Bellingshausen Seas." Geophysical Research Letters 41, no. 22 (November 2014): 7942–49. https://doi.org/10.1002/2014GL061600.

p. 7 Fig 3: "ppt" should be "PSU". Typically, we use degrees C instead of K in cryospheric research but that makes no difference for this particular plot.

p. 7 l. 6-20: Most of this paragraph seems simply to describe Fig. 4 without providing any physical insight into why these differences occur. To a limited degree, it is helpful to have you point out the most salient features of each panel but it would be far more useful to get some understanding of why changes in salinity occur where they do (and similarly for temperature). Why are they so different?

p. 7 l. 13-15: It is not at all obvious to me how you are backing up the assertion that freshwater flux is more significant than heat flux. The way I would expect to see that is in the influence of each on density changes, which in turn affect large-scale overturning and mixing into the deeper ocean. But Fig. 4 provides no information about the effects on density. Given that T and S have completely different units, there seems to be no

basis for comparing the relative importance of these differences on their own. The fact that temperature differences are more scattered does not seem in any obvious way to support the conclusion that heat fluxes are less influential on these differences.

p. 7 l. 19: I think it would make sense to include the figure indicated by "Figure not shown", as I think the changes in the ACC would be an important finding.

p. 7 l. 19-20: The discussion of Fig. 5 is so short that it is not at all clear what the figure is justified. I did not get any physical insight into the spatial pattern of freshening at the seafloor from the figure or the discussion here.

p. 8 Fig 4: The axis need descriptive labels including units. Tick mark labels should be larger. Caption should include the color of the curves (since figure will always be in color). There is no obvious reason that the x axes of the 3 panels are different, and this makes comparing the panels more difficult. The x axis is for both salinity and temperature differences? The depth axis should be inverted so that the deep ocean is down. It is also standard to have these depths be negative, indicating that they are elevations below sea level. What is the northern boundary of each of these regions? What longitudes separate them?

p. 9 Fig 5: This figure does not seem at all useful to me. The color contours are set such that all we can tell is the sign of the salinity difference (and that it is greather than -0.05 PSU) over the vast majority of the sea floor. A nonlinear color bar or one with many more contour values would be needed to make this figure at all useful.

p. 9 l. 5 "surface ocean": A careful point has been made in the manuscript that the freshwater flux is not at the ocean surface, so this should probably be "upper ocean".

p. 9 l. 5-p. 10 l. 2: This paragraph again refers to "anomalies", whereas I would encourage you to use "differences". Other than this small issue, I think this paragraph has some of the best analysis in the paper.

p. 9 l. 9-10: It's not clear to me what the difference between the warm advection

anomaly and the sarm SST anomaly is. It seems obvious that the one would cause the other but maybe I'm missing something.

p. 9 l. 10-11: "The cold water from BMR is advected by the ACC westward": A couple of things here, the ACC flows eastward (which seems to be the direction most of the cold difference is being advected) not westward. There is also the Antarctic Coastal Current (ACoC) that does flow westward on the continental shelf so maybe that's what is advecting a bit of the colder melt water to the west toward that SIT dipole?

p. 9 l. 15: I don't understand the cause of the increased SST near the sea-ice edge. Could you explain further why downwelling is associated with increased SST?

p. 9 l. 16: It seems worth exploring in more detail *how* the results from the two studies are different, not just to point out that they are different and that they are simulating different conditions (transient vs. quasi-steady; cavity geometry vs. no cavities for the "control").

p. 10 Fig 6: labels (tick marks, lat/lon, color bars) are all far too small. Please make them bigger an crisper. Please add more lines for lat and lon if possible so the reader can more easily find the lat/lon coordinates identified in the text.

p. 11 l. 9-10: It seems entirely backwards to me that including basal melting would decrease the MOC. If the model were correctly producing more AABW from ice-shelf melting and subsequent climate and topographic interactions, there should be an increase in downwelling just off the Ross continental shelf break and an associated increase in southward transport in the upper ocean (by conservation of mass). This should lead to an increased MOC strength. This is my understand of the main contribution of Antarctic climate dynamics to the global ocean circulation. To me, the decreased MOC in your simulations with melt fluxes suggest that something is wrong in the simulations and AABW is not being produced. This would not be surprising at coarse resolution, since ESMs have a very hard time producing AABW for the right reasons at CMIP-type resolutions.

p. 11 l. 13: The formation and spreading of AABW should be the cause (not the effect) here. Changes in AABW formation should be driving the changes in the MOC.

p. 12 l. 4-6: I suggest you look further into these difference as part of this paper. It is precisely this kind of comparison with previous work that I feel is missing from this paper. Without more of this kind of validation work, it remains hard to trust the conclusions about the effects of melt fluxes on the ocean-sea ice system.

p. 12 Fig. 9: I find it very hard to tell what is going on with the difference contours. The color plot is quite clear in most regions but hard to discern near the Antarctic the contours are hard to get the sign of, let alone the magnitude in the Antarctic. Maybe the figure should give more space to the region from -90 to -60 (i.e. a nonlinear x axis).

p. 12 l. 14: "contributes to northward heat transport anomaly": I find this confusing, since at least in the real world there should be a consistent southward transport of heat. In your simulations, you seem to see a mix of northward and southward transport do the "anomaly" is contributing to a reduction in southward heat transport at some latitudes and enhanced northward transport in others. Maybe "contributes to a reduction in southward heat transport"?

Also, this needs some discussion. Consistent with my concern about the MOC above, it seems like you should be seeing steady southward heat transport in both cases and that southward heat transport should be enhanced by AABW formation, whereas you are seeing a consistent global reduction (with varying behavior in each ocean basin).

The discussion of the individual basins is clearer in terms of describing enhanced or reduced transport.

p. 13 Fig 10: It would be helpful to compare the global MHT in 10a with observations, such as: Trenberth, Kevin E., and Julie M. Caron. "Estimates of Meridional Atmosphere and Ocean Heat Transports." Journal of Climate 14, no. 16 (August 1, 2001): 3433–43. https://doi.org/10.1175/1520-0442(2001)014<3433:EOMAAO>2.0.CO;2.

Eyeballing the comparison, the global MHT isn't too bad north of 40S but it is odd that you are seeing significant *northward* transport of heat between 60S and 40S, which is not consistent with observations.

p. 13 Conclusion and discussion: Overall, I find that this is mostly just a summary of the results with insufficient interpretation of the findings, discussion of the implications of this work for other modeling efforts and/or the behavior of the "real world" and insufficient introspection about what the missing processes and other shortcomings of the work might be.

p. 13 l. 9: "profoundly": This is a very subjective term and I'm not sure it is supported by the results. The differences between simulations with and without RIS melting are detectable to be sure but the changes generally seem to be subtle rather than profound.

p. 13 l. 9-p. 14 l. 2: My concerns about the "latent heat flux anomaly" and associated complexity of the temperature evolution remain the same as above. I do not think there has been sufficient analysis of the physical processes leading to the temperature evolution to conclude that they are even the result of the latent heat flux from ice-shelf melting. Instead, they are likely to result primarily from density changes, which are in turn primarily controlled by freshwater fluxes. Thus, I think the conclusion that the latent heat flux plays a secondary role is correct but I don't think anything presented in this manuscript has supported that conclusion directly.

p. 14 l. 3-9: The manuscript did not present the circulation from either EI or EN or make any attempts to compare these with observations, so it is difficult to know how much (if any) credence can be given to the difference in circulation between the two experiments. That being said, Again I find the discussion of the surface processes to be among the most useful analysis in the paper.

p. 14 l. 7: Again, the fact that basal melting stabilizes the water column and weakens overturning just seems to indicate that the processes we know to occur as part of AABW formation are missing from the model.

p. 14 l. 10-13: The discussion of fixed ice-shelf area seems unrelated to the manuscript and its findings. There is nothing to suggest that having dynamic ice-sheet geometry in this configuration would enhance our understanding of the quasi-equilibrium state of the ice sheet-ocean-sea ice system because: 1) the resolution of the ocean model is very much insufficient to supply realistic melt patterns to drive ice-sheet evolution; 2) the steady-state melting, if consistent with present-day observed melting, would be unlikely to drive any significant ice-sheet evolution because melt rates under RIS are very small. 3) the context in which melt-driven ice sheet dynamics are interesting are precisely those that are *not* in quasi-equilibrium.

p. 14 l. 14-19: I appreciated this discussion of possible future directions for the research.

Typographical and Grammatical Corrections:

Title: The title would read better as "Modelling the effect of Ross Ice Shelf melting on the Southern Ocean in quasi-equilibrium"

p. 1 l. 6: "basal melting of Ross Ice Shelf" should be "basal melting of *the* Ross Ice Shelf"

p. 1 l. 19: remove "And, ". It is not necessary and is grammatically incorrect.

p. 1 l. 20: "accompanied accordingly": This phrase doesn't make sense. Perhaps you mean something like, "There is an accompanying northward anomaly in meridional heat transport at most latitudes of the global ocean"?

p. 1 l. 23: "Ices accumulated . . . are" should be "Ice accumulated . . . is". Ice is only plural if there are multiple classes of ice or something along those lines, which doesn't seem to be the case here.

p. 2 l. 11: 2 should be written out a "two".

p. 2. l. 13: I suggest changing "regarding" to "of".

p. 2 l. 17: here and elsewhere "sub-ice shelf" should be "sub-ice-shelf"

p. 2 l. 19: "representation" should be "representations"

p. 2 l. 21: "parameterization should be "parameterized"

p. 2 l. 27: "In study such as modeling Ice Shelf melting effect on the Ocean…" this whole sentence is needs some significant grammatical work. Here's my best guess at what is intended: "In studies that include the effect of ice-shelf melting on the ocean in quasi-equilibrium, it is necessary to use a global model with thermodynamically active ice-shelf cavities and to perform integration over hundreds of years"

p. 3 l. 13-14: "assuming the RIS being in steady state" should be "assuming the RIS to be in steady state"

p. 3 l. 17: "should be "to get \*the\* RIS draft"

p. 3 l. 29 and 31: "1ooo" should be "1000" (zeros, not o's).

p. 3 l. 29: "To resolve the RIS vertically better" should be "To better vertically resolve the RIS"

p. 4 l. 3: "to that in" should be "to those in"

p. 4 l. 8-9: "and the Antarctic situates on the.." should be "with Antarctica situated on the…"

p. 4 l. 9: "the bathymetry of ocean around the Antarctica and cavity geometry of RIS is" should be "the ocean bathymetry around Antarctica and the cavity geometry of RIS are"

p. 4 l. 10: "grids" should be "grid cells" or "grid boxes". To me, the whole 64x64 face is a grid.

p. 4 l. 11: "of which 15 having cavities and being calculated basal melting" should be something like "of which 15 have nonzero cavity thickness and include basal melt

calculations"

p. 4 Fig. 1: "(a)" and "(b)" should go before the phrases describing each panel rather than after.

p. 4 Fig. 1: "yellow shades in (b) indicate grids where cavities..." should probably be "grid boxes shaded light green indicate locations where cavities...". (To my eyes, the shading is light green, not yellow.)

p. 5 l. 15: "modelling ice shelf" should be "modeled ice shelf". "lateral boundary" should be "lateral boundaries".

p. 5 Fig 2: The tick mark labels on the axes are too small to easily read. The melt-rate values are also somewhat small but perhaps large enough to read (but I see no reason to include so many empty cells around the 15 active cells. The 3 panels probably will need to be combined into a single figure for typesetting but I guess that's up to you and the journal to work out.

p. 5 Fig 2: "annual mean areal average" should probably be something like "the annual and area mean"; "for the last 100 years' mean" should be "for the mean over the last 100 years"; "areal mean averaged over the last 100 years" might be clearer as "averaged over the ice-shelf area and the last 100 years".

p. 6 l. 6: "cold and fresh water are" should be "cold and fresh water *is*"

p. 6 l. 7: "become" should be "becomes"

p. 6 l. 7: "compared its counterpart" should be "compared *to* its counterpart"

p. 6 Fig 3: The axis labels should be more descriptive (not variable names) and should include units.

p. 7 l. 1: "Figure 3. Figure 3." should just be "Figure 3"

p. 7 l. 6-20: There is no need to continually reference Fig. 4 here. It is clear that most

of this text refers to that figure.

p. 7 l. 7: "from ocean surface" should be "from \*the\* ocean surface"

p. 7 l. 9: "freshening effect" should be "\*the\* freshening effect"

p. 7 l. 16: why 1005 m instead of just 1000 m?

p. 7 l. 17-18: "This is due to that": this phrase is kind of confusing. I would suggest something like "This is due to the relatively stronger...at that level, which constrains..."

p. 7 l. 19: "water in most area" should be "water in most of the area"

p. 9 l. 16 and p. 10 l. 1: "the work" should be "this work"

p. 10 l. 12: "BMR effect" should be "the BMR effect" (or maybe "the Ross BM effect", see earlier comment).

p. 11 l. 1: No need to reference Fig. 8 again.

p. 11 l. 2: "Figures not shown" should just be "not shown"

p. 11 l. 3: "motion field" should be "flow field"

p. 11 l. 8: "by meridional transport" should be "by a meridional transport" or "by the meridional transport"

p. 11 l. 10 "here" should be lowercase or this should be made a separate sentence in parentheses (though The Cryosphere's typographic editors discourage theses)

p. 11 l. 15 "the path" should be lowercase

p. 11 l. 15 "it's" should be "it is"

p. 12 l. 2 "the calculation" should be lowercase

p. 12 Fig. 9: Axes need labels including units. All labels are too small to be readable.

p. 12 l. 14: "contributes to northward" should be "contributes to the northward"

p. 13 Fig 10: the customary way of handling multiple y axes is to put one axis on the left of the figure and the other on the right. It is even more helpful if the axes are the same color as the curves they correspond with. As in all figures, the tick mark labels are far too small.

p. 14 l. 7: "stables" should be "stabilizes"

---

## Author Comment (AC1) · 8 Mar 2018

I want to thank R. Walker for his constructive review and good suggestions. I am answering his comments in the following. For clarity, I repeat the original comment ([C]) at first and then the answer ([A]) and author's changes in manuscript [R] afterwards: General Comments

[C]: Inclusion of ice shelves in global circulation models is a significant issue for the accuracy of climate projections. This study considers the impact of basal melting under the Ross Ice Shelf on the Southern Ocean by contrasting global ocean model experiments with and without melting in the sub-Ross cavity. The choice of a no-melt scenario that includes sub-ice-shelf bathymetry seems a little odd to me, as most ocean modeling that I'm aware of either includes ice shelves plus melting or excludes ice shelves from the domain. It should still be possible to get value from this experimental setup. However, I would have liked this manuscript to spend much more time on detailed discussion of the different experiments, particularly the relations between water properties and dynamics.

[A]: Initially I set up two experiments, one included ice shelves plus melting and the other excluded ice shelves from the domain. After preliminary analysis of simulation results, I realized that the sub-ice-shelf bathymetry gave significant contribution to the differences between the results from the two simulations. This difference in geometry changes local circulation and mixing and leads to changes of overall results compared to or even greater than that in basal melting under the Ross Ice Shelf. Under such conditions, it would be difficult to discuss the effect of basal melting under the Ross Ice Shelf. Hence a third experiment with no-melt scenario that includes sub-ice-shelf bathymetry was added and its results were used in the discussion instead of that from the experiment that excluded ice shelves from the domain. More discussions on the modelling results have been added.

[R] See [R] parts for specific comments.

[C]: General comment on figures) All units should be in axis labels, not only in the captions. Also, axes should be labeled with variable names. Figures 1, 2, 6, 7, 9, 10 should have a larger font size to be readable.

[A&R] These figures have been redrawn:

---

## Author Comment (AC2) · 8 Mar 2018

I want to thank X. Asay-Davis for his thorough review and good suggestions. I am answering his comments in the following. For clarity, I repeat the original comment ([C]) at first, then the answer ([A]) and author's changes in manuscript [R] afterwards:

General Comments

[C] This paper describes two global ocean-sea ice experiments run to quasi-equilibrium over 500 years, one with basal melting below the Ross Ice Shelf (RIS) and one without basal melting (but still with an ocean cavity below RIS). Most of the presented results examine differences between ocean and sea ice properties between the two experiments (results are typically averaged over the last 100 years of each experiment). Non-negligible differences in the horizontal and vertical distributions of temperature and salinity are found between the two experiments, leading to appreciable differences in both the meridional overturning circulation and meridional heat transport in the ocean. Differences in the surface and barotropic flow are also demonstrated, along with related changes in sea surface temperature, sea ice thickness and sea ice concentration.

The purpose of the study seems to be to show that freshwater fluxes from ice shelves have a significant impact on the Southern Ocean compared to a simulation without any freshwater fluxes. It seems unclear (and is not discussed in the manuscript) what the relevance of this study is to other modeling work or what this study might tell us about the melt-induced dynamics in the real world.

[A] The purpose of the study is to show the impacts of freshwater and latent heat fluxes from basal melting of Ross Ice Shelve on the Southern Ocean compared to a simulation without any basal melting. More discussions on the relevance of this study to other modeling work and on modeling results have been added.

[R] See the [R] part for specific comments.

[C] It has been known for some time in the Earth System Modeling community that some form or freshwater input into the deep ocean is required for adequate representation of deep ocean properties and of the meridional overturning circulation. Therefore all global Earth System Model (ESM) include some mechanism for freshwater input (typically surface "runoff" around Antarctica and Greenland), together with a mechanism for inducing overturning in polar regions (typically salinity restoring at the surface). No models I am aware of without sub-ice-shelf melting would leave out these mechanisms. Therefore, if the aim of this study is to show that current ESMs should be including the effects of ice-shelf melting in order to avoid inaccuracies in Southern Ocean properties, the "control" experiment (EN in the manuscript) should probably have been closer to a configuration used in ESMs: "runoff" at the surface to at least partially account for freshwater fluxes and no ice-shelf cavities.

[A] The numerical model of any kind used in scientific studies is a simplification and approximation of the real world, no matter how rough or elaborate it is. As have been pointed out in the review, in global ocean modelling the "runoff" can be tailored to reflect the freshwater input connected with basal melting of ice-shelf basing on assumptions. In this way, the latent heat flux is ignored. In addition, the difference in bathymetry with and without ice-shelf cavities changes local circulation and mixing and leads to changes of overall results compared to or even greater than that in basal melting under the Ross Ice Shelf. In the work, the "run off" is not used. The aim of this study is to investigate what differences may be led by including the effects of

ice-shelf melting under the Ross Ice Shelf to Southern Ocean properties. For this aim, the bathymetry is identical in the two experiments.

[R] See the [R] part for specific comments.

[C] If the purpose of the study is to show what features of the climate system are affected by the presence or absence of melting below RIS, there is another significant pitfall in this work. Very little effort is made to validate either the EI (with melting) or the EN (without melting) experiment against observations or previous modeling (except for the basal melt rate below RIS). This strikes me as highly problematic because the differences between the simulations is unlikely to tell us something about the real world if the base state (either EI or EN) that is being perturbed can be shown to be representative of the real world. Given the *very* coarse horizontal resolution (150 km) and rather coarse vertical resolution (not stated but seemingly around 50 m), it seems unlikely that the model will be able to capture the complex chain of processes by which water masses are transformed on the Ross continental shelf, within the Ross cavity, and off the continental slope where they mix into the deep ocean. These processes have been shown to require horizontal resolutions at least 30x higher than this simulation (see specific comments), allowing interactions between small-scale topographic features and narrow oceanic currents. Without these transformations being captured adequately or the model state having been validated against a broader set of observations, conclusions in this work about how basal melting affects the Southern Ocean are likely to only apply to this particular model configuration, and not to be representative of the real world.

[A] More efforts have been made to validate the EI experiment against previous (for example, heat transport). The choice of model resolution is determined by the problem to study. The purpose of the study is to show what features of the climate system are affected in large scale by the presence or absence of melting below RIS. Under the current resolution, major features of bathymetry of RIS can be resolved and the influence of fresh water flux and latent heat flux due to basal melting of RIS can be represented. The influence of sub-grid processes on modeling results needs further study. Whether the conclusions in this work are model dependent or not also needs further work in the community. To approach the true result, more modelling work with different models are needed. Even most models give similar result, it is still possible that the result is not representative of the real world. Discussion on this have been added in the manuscript.

[R] See the [R] part for specific comments.

[C] The manuscript presents much of the results results with little deeper analysis, discussion or explanation (the exception is a more careful analysis changes in sea surface temperature and sea ice properties resulting from flow anomalies near RIS). Except for the dynamics at the ocean surface, little attempt is made to explain how water masses are transformed to reach various ocean depths. Basal melting is found to *decrease* the global overturning circulation, seemingly due to increased stabilization of the water column, in contradiction to know physical processes of Antarctic Bottom Water formation (known to occur in the Ross Sea region) that are thought to be an important driver of global ocean circulation. No discussion is included of potential shortcomings of the model at capturing or resolving ocean processes that would be relevant to these transformations.

I can only recommend this manuscript for publication after major revisions to address these shortcomings.

[A] More analysis and discussion on the results have been added in the revision version. AABW is formed in the Southern Ocean from surface water cooling in polynyas. With basal melting effect included, sea ice concentration in the Ross Sea increases and more salts are rejected to the ocean. Due to the adoption of boundary condition of restoring salinity in the simulations, the sea surface salinity increase from more ice freezing cannot be reflected in the model. Increased basal melting changes the shelf water characteristics and increases the stability of the water column, decreasing deep convection and the formation of denser bottom water (Hellmer, 2004). In the study of Kusahara and Hasumi (2013), it is found that if the basal melting of ice shelves is included, weakening of the thermohaline circulation driven by Antarctic dense water formation under warming climate conditions will be enhanced. During preparing the manuscript, I have tried to explain how water masses are transformed to reach various ocean depths. I inspected the time series of area-averaged difference in salinity and temperature at different levels for southern Atlantic Ocean, southern Pacific Ocean and southern Indian Ocean respectively. I also analyzed the lead/lag correlations between the fresh water flux from RIS basal melting and salinity of the Southern Ocean. But I have not got a clear picture. Since the time-dependent virtual tracers in the oceans can provide information on the ocean circulation, it would be a better way to make use of tracers to estimate pathways in the ocean.

[R] See the [R] part for specific comments.

[C] This paper would benefit from significant editing by a native English speaker. I have attempted to point out typos and grammatical errors where I have seen them (I include about 3 pages of such corrections). Additionally, the figures all need significant format-ting work before they are ready for publication, including labeling axes and increasing font sizes to make the labels more readable.

[A] Thanks so much for correcting the errors in language usage which have all been accepted. The revised manuscript will be edited by a native English speaker from Editage, a company supplying language services. All figures have been redrawn to meet the demand for publication.

[R] See the [R] part for specific comments.

Specific Comments

[C] p. 1 l. 6: In the field, BMR is typically used as an abbreviation of "basal melt rate". The incorporation of the Ross Ice Shelf into this abbreviation is confusing. I would suggest replacing "basal melting of Ross Ice Shelf (BMR)" with "basal melting below the Ross Ice Shelf (Ross BM)" and elsewhere replace "BMR" with "Ross BM" to avoid confusion. If you can come up with an alternative shorthand that will not be confused with "basal melt rate", that would be fine, too.

[A] The suggestion is accepted.

[R] The abbreviation "BMR" is replaced by "BMRIS" in the revised manuscript.

[C] p. 1 l. 12: I would suggest replacing "substantially" and "not so significant" with something more quantitative if possible.

[A] The suggestion is accepted

[R] The sentence has been modified to "The extra freshwater flux decreases the salinity from 1500 m to the sea floor in the southern Pacific Ocean and the southern Indian Ocean with a maximum difference of nearly 0.005 psu in the Pacific Ocean whereas the effect of concurrent heat flux is mainly confined to the middle layer of water body (roughly from 1500 m to 3000 m)"

[C] p. 1 l. 14: "local circulation anomalies": In general, the abstract seems to treat the case of no basal melting as the control case and the case with basal melting as the modified experiment. I can understand this choice, since ocean models typically do not include ice-shelf cavities, though it seems strange from a physical standpoint to treat the less physical experiment as the control case. Here, the use of the word "anomalies" seems particularly strange to me, since it seems to imply "something that deviates from what is standard, normal, or expected", whereas I would say the control case is the one more likely to deviate from the physical world. Perhaps another phrase such as "differences in local circulation) would be clearer.

[A] The suggestion is accepted.

[R] The "anomalies" has been used ad little as possible.

[C] p. 1 l. 14: "with the help of ocean bathymetry": This phrase seems rather vague to me. Maybe a better wording would be something like "The decreased density due to the effect of Ross BM, together with interactions with ocean bathymetry, creates local differences in circulation in the…"

[A] The suggestion is accepted.

[R] The sentence has been changed to "The decreased density due to the effect of BMRIS, together with the influence of ocean bathymetry, creates local differences in circulation in the Ross Sea and nearby water"

[C] p. 1 l. 22-24: The audience for The Cryosphere is aware of what ice sheets, ice shelves, icebergs, etc. are so I don't think this level of introduction is necessary.

[A] The suggestion is accepted.

[R] The two sentences have be replaced with "Ice shelf melting, which accounts for 55% of the ice mass loss from Antarctica, is one of the main sources of freshwater input to the Antarctic coastal ocean (Mathiot et al.,2017)".

[C] p. 1 l. 26: "beneath the currently stable Ross Ice Shelf": The phrase "currently stable" is both grammatically problematic and confusing, because it implies a past or future instability in RIS that is not addressed here, nor is there any widely accepted likelihood of RIS instability in the community. I would remove this phrase.

p. 1 l. 26: "can be larger than 2500% of the overall: : :": It is not clear that this fact or this reference is relevant to the rest of the paper, as you are not resolving melt channels in your simulations.

[A] The suggestions are accepted.

[R] The sentence has been removed.

[C] p. 2 l. 3: "Neglecting the sub-ice freshwater...for the Southern Ocean." While it is not stated here, the implication seems to be that common practice in ocean modeling of the Southern Ocean is to neglect sub-ice-shelf freshwater fluxes entirely, whereas this is not usually the case. Global (and I believe also regional Antarctic) ocean models without ice-shelf cavities still include an approximation of the total Antarctic freshwater input (melting + calving) but they almost universally do so by distributing the freshwater at the ocean surface and typically evenly around the continent. In my view, sub-ice-shelf freshwater fluxes aren't really "neglected" so much as they are estimated and distributed inaccurately. Here is one publication that discusses the differences in ocean model behavior depending on how freshwater fluxes are distributed: Mathiot, P., Jenkins, A., Harris, C., and Madec, G.: Explicit representation and parametrised impacts of under ice shelf seas in the z* coordinate ocean model NEMO 3.6, Geosci. Model Dev., 10, 2849-2874, https://doi.org/10.5194/gmd-10-2849-2017, 2017.

[A] To avoid misunderstanding, the sentences have been modified.

[R] The sentences has been changed to "The sub-ice freshwater input has various implications for the Southern Ocean."

[C] p. 2 l. 3-4: "These are pronounced in the Weddell...broad continental shelves". It is not clear to me that the Weddell and Ross Seas are the regions of Antarctica that would be most affected by neglecting freshwater fluxes. The large size of RIS and Filchner-Ronne Ice Shelf (FRIS) together with their relatively cold ice-shelf cavities do make them particularly important for AABW formation but other regions of Antarctica with warmer cavities have been shown to produce significant amounts of freshwater that impact Antarctic climate both locally and regionally in significant ways, see e.g.: Nakayama, Y., R. Timmermann, C. B. Rodehacke, M. Schröder, and H. H. Hellmer (2014), Modeling the spreading of glacial meltwater from the Amundsen and Bellingshausen Seas, Geophys. Res. Lett., 41, 7942–7949, doi:10.1002/2014GL061600. The Getz, Thwaites and Pine Island Ice Shelves, for example, each produce significantly more freshwater than RIS and nearly as much as FRIS, despite their significantly smaller areas: Rignot, E., Jacobs, S., Mouginot, J., Scheuchl, B. Ice-shelf melting around Antarctica. Science. 2013 Jul 19;341(6143):266-70. doi: 10.1126/science.1235798.

[A] To resolve the smaller ice-shelves, model resolution must be improved greatly. Besides, the scenario would be different from the work if smaller ice-shelves are included.

[R] Discussion has been added in Part 4: " Ice shelves range in size from 500 000 km^2 (RIS) to around 100 km^2 (Ferrigno ice shelf) . The current global ocean model configurations cannot resolve explicitly all the ice shelf cavities, especially for large scale simulation. As have been illustrated by some studies (for example, Rignot et al., 2013; Nakayama et al.,2014), small Ice Shelves can produce significantly more freshwater than RIS and impact Antarctic climate both locally and regionally in significant ways. Not all Ice Shelves are in stable state (some are thickening and some are thinning) (Rignot et al., 2013). To study the influences of stable Ice Shelf basal melting on the Southern Ocean in the long run, the RIS is included under the affordable model resolution for a long integration. But a model's horizontal resolution is important not only in simulating the conditions underneath the ice shelf that lead to basal melt but also for the conditions in the open ocean that deliver heat to ice shelf cavities (Dinniman et al.,2016).

Increasing the model resolution dramatically improves the representation of Circumpolar Deep Water on the Amundsen Sea continental shelf (Nakayama et al., 2014; Dinniman et al., 2015). So more work with finer resolution should be carried out to reduce the uncertainty in simulation of BMRIS effect on the Southern Ocean. Besides, the effects of other ice shelves, such as the Filcher-Ronne and so on, should also be evaluated."

[C] p. 2 l. 8-9: It would be good to supply a more complete list of estimates of basal melting. Here are a few more important ones:
Moholdt, G., L. Padman, and H. A. Fricker (2014), Basal mass budget of Ross and Filchner-Ronne ice shelves, Antarctica, derived from Lagrangian analysis of ICESat altimetry, J. Geophys. Res. Earth Surf., 119, 2361–2380, doi:10.1002/2014JF003171.
M. Depoorter, J. Bamber, J. Griggs, J. Lenaerts, S. Ligtenberg, M. van den Broeke, G. Moholdt. Calving fluxes and basal melt rates of Antarctic ice-shelves. Nature, 502 (7469) (2013), pp. 89-92
[A] The suggestion is accepted.
[R] Result from Moholdt et al. (2014) has been added in the manuscript.

[C] p. 2 l. 10: Other sources (Rignot et al 2013, Depoorter et al. 2013) estimate a significantly larger mean melt rate on the order of 0.8-0.9 m/a. Beckmann and Goosse, 2003 is not really an appropriate citation for the 0.5 m/a number, they are merely citing the Jacobs et al. 1996 estimate, converted from mSv to m/a. Given the significant improvements in satellite observations since 1996, I do not feel that number is particularly trustworthy.
[A] I agree.
[R] The number from Rignot et al 2013 has been used in the revision.

[C] p. 2 l. 11: "occurs at the base of the ice shelf edge": This is sometimes true, particularly for warm ice-shelf cavities. But the freshwater plume in cold cavities typically reaches neutral buoyancy at depths significantly below the ice-shelf edge:
Jacobs, S S, H H Helmer, C S M Doakea, A Jenkins, and R M Frolich. "Melting of Ice Shelves and the Mass Balance of Antarctica." Journal of Glaciology 38, no. 130 (1992): 375–87. doi:10.3198/1992JoG38-130-375-387.
For the purposes of the point you are making, it would be sufficient to say, "Since the injection of this freshwater occurs at depth rather than at the ocean surface..."
[A] The suggestion is accepted.
[R] The sentence has been revised as suggested.

[C] p. 2 l. 16-17: "can provide no direction information about sub-ice shelf circulation": This is not entirely true, as sub-ice-shelf observations include velocity measurements that can be used to infer at least some basic information about the sub-ice-shelf circulation. Temperature and salinity measurements can also be used to infer, through the fraction of Ice Shelf Water, the degree of interaction with the ice-shelf base, which also can provide information about the broad sub-ice-shelf circulation. I would suggest toning this down to say that it is difficult to infer the sub-ice-shelf circulation from borehole observations.

[A] The suggestion is accepted.

[R] The sentence has been revised as suggested.

[C] p. 2. l. 15-29: The citations in this paragraph seems out of date and incomplete. These reviews provide many citations that could help to fill in the gaps:

Dinniman, Michael, Xylar Asay-Davis, Benjamin Galton-Fenzi, Paul Holland, Adrian Jenkins, and Ralph Timmermann. "Modeling Ice Shelf/Ocean Interaction in Antarctica: A Review." Oceanography 29, no. 4 (December 2016): 144–53. https://doi.org/10.5670/oceanog.2016.106.

Asay-Davis, Xylar S., Nicolas C. Jourdain, and Yoshihiro Nakayama. "Developments in Simulating and Parameterizing Interactions Between the Southern Ocean and the Antarctic Ice Sheet." Current Climate Change Reports, October 24, 2017, 1–14. https://doi.org/10.1007/s40641-017-0071-0.

I would suggest a complete rewrite of the paragraph with a more complete list of the numerical methods, domains, time periods covered, etc.

In particular, there are several studies that have used the MITgcm with ice-shelf cavities in regional configurations to study Antarctica and the Southern Ocean. Since these use the same model as this study, it would seem like they might get particular emphasis here.

[A] The suggestion is accepted.

[R] The paragraph has been rewritten as: "The need for numerical modelling of ice shelf–ocean interaction is particularly acute due to a lack of extensive observational data, which results from the physical inaccessibility of the areas of interest. Besides, it is difficult to infer the sub-ice-shelf circulation from borehole observations, leaving a significant role to be played by numerical models (Walker and Holland, 2007; Dinniman et al.,2016). As illustrated in Table A1, in ice shelf-sea ice-ocean coupled modelling, researchers use different kinds of ice shelf representation, such as dynamic (Grosfeld and Sandhager, 2004), simplified and computationally inexpensive but capable of handling significant changes to the shape of the sub-ice shelf cavity as the shelf profile evolves (Walker and Holland, 2007), fixed cavity and thermodynamics (Losch, 2008; Timmermann et al., 2012), and parameterization (Beckmann and Goosse, 2003). The models are mostly circumpolar (Hellmer, 2004; Kusahara and Hasumi, 2013; Mathiot et al., 2017), regional (Galton-Fenzi et al., 2012) or two-dimensional in yz-plane (Walker et al., 2009). Beckmann and Goosse (2003) studied the ice shelf basal melting effect using a global ocean-sea ice coupled model with parameterized ice shelf basal melting. Losch (2008) introduced ice shelves into the MITgcm and carried out ISOMIP (Ice Shelf–Ocean Model Intercomparison Project) experiments and nearly global (excluding the Arctic Ocean) ocean circulation experiments. In the nearly global ocean circulation experiments, results with and without explicit modelling of ice shelf cavities were presented and analysis were mainly focused on the Weddell Sea and the circulation in the Filchner-Ronne Ice Shelf cavity. Timmermann et al. (2012) presented results of ice shelves basal mass loss from a global sea ice-ice shelf-ocean model based on the finite element method, in which the model was forced with daily data from the NCEP/NCAR reanalysis for the period 1958-2010. There are also many modelling works on ice shelve in recent years using regional, circumpolar or regional configuration models mentioned in Table A1. Asay-Davis et al. (2017) have given a thorough review on those works. However, in study such as modelling Ice Shelf

melting effect on the Ocean in quasi-equilibrium, using global model with ice shelf thermodynamics for basal melting and performing integration over hundreds of years is of necessity. This kind of work has not been done before.

Table A1. An incomplete list of ice shelf-ocean coupled modelling

| Publication | Ocean model | ice shelf implementation | domain and time periods covered |
|---|---|---|---|
| Beckmann and Goosse (2003) | Bremerhaven Regional Ice Ocean Simulations (BRIOS) | Parameterization | Circumpolar, 100 years |
| Grosfeld and Sandhager, (2004) | a rigid-lid, hydrostatic primitive equation model, formulated in spherical coordinates | Dynamic | 900km x 700km in the horizontal, 300 years |
| Hellmer (2004) | Bremerhaven Regional Ice Ocean Simulations (BRIOS) | fixed cavity and thermodynamics | Circumpolar,20 years |
| Walker and Holland (2007) | A two-dimensional model in the yz-plane | simplified dynamic | 600km x 1100m, 600 years |
| Losch (2008) | MIT general circulation model (MITgcm) | fixed cavity and thermodynamics | In ISOMIP (Ice Shelf–Ocean Model Intercomparison Project) experiment: from 0ºE to 15ºE and 80ºS to 70ºS, 10 000 days In (nearly) global ocean model (excluding the Arctic Ocean) experiment: 80ºN southward, 100 years |
| Timmermann et al. (2012) | Finite Element Sea-ice Ocean Model (FESOM) | fixed cavity and thermodynamics | Global, 53 years |
| Galton-Fenzi et al. (2012) | Regional Ocean Modeling System (ROMS) | fixed cavity and thermodynamics | Regional, 20 years |
| Kusahara and Hasumi (2013) | a sea ice-ocean coupled model, named COCO | fixed cavity and thermodynamics | Circumpolar,25 years for CTRL run and 38 additional years for ERA-INT case |
| Mathiot et al. (2017) | Nucleus for European Modelling of the Ocean (NEMO) | fixed cavity and thermodynami | In academic case: from 0ºE to 15ºE and 80ºS to 70ºS, 10 000 days In real ocean application: circumpolar, |

| | cs | 10 years |
|---|---|---|

"

[C] p. 2 l. 19: "dynamic": this could use further clarification. I think you mean dynamic ice-shelf geometry? How is this different from Walker and Holland (2007)?

[A] Yes, I mean dynamic ice-shelf geometry. Walker and Holland (2007) scheme is simpler and only permits one-dimensional flow.

[R] It has been revised to "dynamic ice-shelf geometry permitting two-dimensional flow"

[C] p. 2. l. 21: "fixed cavity and thermodynamics": The cavity geometry is fixed but the thermodynamics is not – melt rates evolve with changing ocean conditions.

[A] Thanks for pointing out the problem.

[R] "fixed cavity and thermodynamics" has been revised to "thermodynamics with fixed cavity".

[C] p. 2 l. 21: "parameterization": Again, more details on what this means would be helpful.

[A] The suggestion is accepted.

[R] The "parameterization" has been extended to "parameterization of the interaction between ice shelves and the adjacent ocean"

[C] p. 2 l. 22-23: What would the other options be besides the list given? Global? Indeed, there are several studies with global models (Losch, 2008; Helmer et al. 2012; Timmermann et al. 2012, etc.)

[A] Losch, 2008 and Timmermann et al. 2012 were mentioned in the paragraph. Helmer et al. 2012 used a regional model identical to that in Hellmer (2004) which had been mentioned.

[C] p. 2 l. 23: "two-dimensional" needs more clarification – one horizontal dimension and one vertical.

[A&R] "two-dimensional" has been revised to "two-dimensional in yz-plane".

[C] p. 2. l. 28-29: "At present, this kind of research has rarely been reported." I think it is fair to say that this has not been done before.

[A&R] The sentence has been changed to "At present, this kind of research has not been done before".

[C] p. 2 l. 30-p. 3 l. 6: Again, I think this paragraph is missing some important work. Many modeling efforts not mentioned here include the Ross Sea in larger regional or global models that are big enough to look at the effect of RIS on the Southern Ocean. Two examples are:

Timmermann, Ralph, and Hartmut H. Hellmer. "Southern Ocean Warming and Increased Ice Shelf Basal Melting in the Twenty-First and Twenty-Second Centuries Based on Coupled Ice-Ocean Finite-Element Modelling." Ocean Dynamics 63, no. 9–10 (October 2013): 1011–1026. https://doi.org/10.1007/s10236-013-0642-0.

Dinniman, Michael S., John M. Klinck, Eileen E. Hofmann, and Walker O. Smith. "Ef fects of Projected Changes in Wind, Atmospheric Temperature, and Freshwater Inflow on the Ross Sea." Journal of Climate, December 1, 2017. https://doi.org/10.1175/JCLID-17-0351.1.

You are correct that these models were not able to run for long enough times to look at

quasi-equilibrium effects

[A] Thanks for giving the references. Both articles focus on influences of warming atmosphere on Southern Ocean and Ice Shelf. I couldn't find much information on the effect of RIS on the Southern Ocean.

[C] p. 3 l. 12: "will be an interesting topic": I don't think this belongs here, as it is a very subjective statement. I would remove this whole sentence.
[A&R] The statement has been removed.

[C] p. 3 l. 17-19: Both the topography data and the forcing data are not the most up-to-date versions, see references below. Both Bedmap2 (Fretwell et al. 2013) and RTOPO2 (Schaffer et al. 2016) have updated topography, though I am not sure whether these changes affect RIS specifically. There is a CORE-NYF.v2 data set (http://data1.gfdl.noaa.gov/nomads/forms/core/COREv2/CNYF_v2.html), which is a climatology from the interannual forcing described in Large and Yeager (2009). It would be worth explaining why these earlier versions were used instead of the more up-to-date versions.

Fretwell, P, H D Pritchard, D G Vaughan, J L Bamber, N E Barrand, R Bell, C Bianchi,et al. "Bedmap2: Improved Ice Bed, Surface and Thickness Datasets for Antarctica." The Cryosphere 7, no. 1 (2013): 375–93. https://doi.org/10.5194/tc-7-375-2013.

Schaffer, Janin, Ralph Timmermann, Jan Erik Arndt, Steen Savstrup Kristensen, Christoph Mayer, Mathieu Morlighem, and Daniel Steinhage. "A Global, HighResolution Data Set of Ice Sheet Topography, Cavity Geometry, and Ocean Bathymetry." Earth System Science Data 8, no. 2 (October 2016): 543–57. https://doi.org/10.5194/essd-8-543-2016.

Large, W. G., and S. G. Yeager. "The Global Climatology of an Interannually Varying Air–sea Flux Data Set." Climate Dynamics 33, no. 2–3 (August 2009): 341–64. https://doi.org/10.1007/s00382-008-0441-3.

[A] Almost all the work was done in 2015, when the new version cavity geometry dataset hadn't come out. The CORE-NYF.v2 data set was used and now I realized that the reference given in the manuscript was not accurate (on the website, it still says that "Details are provided in the Large and Yeager (2004) report"). There are differences in RIS cavity geometry between RTopo105b and RTOPO2 (Schaffer et al. 2016) and the model reflects these differences in 5 grids with difference of 50 m in thickness of water column in the cavity.
[R] The reference for CORE-NYF.v2 has been revised.

[C] p. 3 l. 17-19: How is "runoff" handled in each experiment (EI and EN)? I believe CORE specifies a runoff field that inputs freshwater into the Antarctic region equally around the continent and at the ocean surface at a level that is supposed to roughly match the surface accumulation over the continent (therefore accounting for the combined effect of runoff, sub-ice-shelf melting and calving, assuming AIS is in equilibrium). Was this runoff field included in your simulations?
[A] This runoff field was not used in both experiments.

[C] p. 3 l. 24-26: I would suggest making this sentence a footnote.
[A&R] It has been moved to footnote.

[C] p. 3. l. 27-28: Please explain the abbreviations "EI" and "EN".

[A&R] The sentence has been revised to "The two experiments are denoted by EI (experiment with basal ice-shelf melting considered) and EN (experiment with no basal ice-shelf melting considered) respectively.

[C] p. 3 l. 29-30: More detail should be given about what the vertical resolution actually is. What is the resolution at the surface? At 1000 m depth? The coarsest resolution (at depth)? I suspect that, even with finer resolution in the upper 1000 m, 30 layers is inadequate to resolve the sub-ice-shelf plume in detail. Finer resolution would likely lead to a significantly different answer, see:

Losch, M. "Modeling Ice Shelf Cavities in a z Coordinate Ocean General Circulation Model." Journal of Geophysical Research 113, no. C8 (August 2008): 1–15. https://doi.org/10.1029/2007JC004368.

Schodlok, Michael P., Dimitris Menemenlis, Eric Rignot, and Michael Studinger. "Sensitivity of the Ice-Shelf/Ocean System to the Sub-Ice-Shelf Cavity Shape Measured by NASA IceBridge in Pine Island Glacier, West Antarctica." Annals of Glaciology 53, no. 60 (2012): 156–162. https://doi.org/10.3189/2012AoG60A073.

Schodlok, M. P., D. Menemenlis, and E. J. Rignot. "Ice Shelf Basal Melt Rates around Antarctica from Simulations and Observations." Journal of Geophysical Research: Oceans 121, no. 2 (February 2016): 1085–1109. https://doi.org/10.1002/2015JC011117.

[A] The layer thicknesses are 10, 10, 15, 21, 28, 36, 45, 13 x 50, 100, 200, 300, 400, 500, 600, 700, and 3 x 800 m. According to Losch (2008), "Dz = 100 m appears to be the minimum vertical resolution that is required to resolve ice shelf-ocean processes." The current vertical discretization meets that standard. The vertical resolution near the bottom is poor. This problem is partially alleviated by the partial cell treatment of topography (Adcroft et al., 1997).

Losch, M.: modelling ice shelf cavities in a z coordinate ocean general circulation model, J. Geophys. Res., 113, C08043, doi:10.1029/2007JC004368, 2008.

Adcroft, A., Hill, C., and Marshall, J.: Representation of topography by shaved cells in a height coordinate ocean model, Mon. Weather Rev.,125(9), 2293 – 2315, doi:10.1175/1520-0493(1997)125<2293:ROTBSC>2.0.CO;2, 1997.

[R] The detail about the vertical resolution has been added.

[C] p. 4 l. 2: "the horizontal resolution is about 150 km". This is one of my biggest concerns about this work. I realize that long time integrations are expensive but this coarse resolution (coarser even than CMIP5 and CMIP6 models of the region) seems *far* too coarse to capture the relevant dynamics for the Antarctic region, most importantly the pathways for transporting freshwater from the RIS to the Southern Ocean. See the following paper for a discussion of the pathways and the resolution (~5 km) required to capture them:

Dinniman, Michael S., John M. Klinck, Eileen E. Hofmann, and Walker O. Smith. "Effects of Projected Changes in Wind, Atmospheric Temperature, and Freshwater Inflow on the Ross Sea." Journal of Climate, December 1, 2017. https://doi.org/10.1175/JCLID-17-0351.1.

See this paper for a discussion of the inadequacy of CMIP5 models at capturing Antarctic continental shelf processes:

Little, Christopher M., and Nathan M. Urban. "CMIP5 Temperature Biases and 21st Century Warming around the Antarctic Coast." Annals of Glaciology 57, no. 73 (September 2016): 69–78. https://doi.org/10.1017/aog.2016.25.

[A] In the history of numerical simulation, coarse resolution modelling was performed before finer work. As mentioned in the review, there are shortcomings in numerical modelling if the resolution is not capable of capturing critical processes. In my opinion, the current configuration is enough for capturing fundamental processes in large scale relating the effect of basal melting of RIS on the Southern Ocean. Smaller ice shelves are not studied in the manuscript.

[R] A short discussion has been added: "Ice shelves range in size from 500 000 km^2 (RIS) to around 100 km^2 (Ferrigno ice shelf). The current global ocean model configurations cannot resolve explicitly all the ice shelf cavities, especially for large scale simulation. As have been illustrated by some studies (for example, Rignot et al., 2013; Nakayama et al.,2014), small Ice Shelves can produce significantly more freshwater than RIS and impact Antarctic climate both locally and regionally in significant ways. Not all Ice Shelves are in stable state (some are thickening and some are thinning) (Rignot et al., 2013). To study the influences of stable Ice Shelf basal melting on the Southern Ocean in the long run, the RIS is included under the affordable model resolution for a long integration in the work. But a model's horizontal resolution is important not only in simulating the conditions underneath the ice shelf that lead to basal melt but also for the conditions in the open ocean that deliver heat to ice shelf cavities and identifying relevant water masses (Dinniman et al.,2016; Little and Urban, 2016). Increasing the model resolution dramatically improves the representation of Circumpolar Deep Water on the Amundsen Sea continental shelf (Nakayama et al., 2014; Dinniman et al., 2015). More work with finer resolution should be carried out to reduce the uncertainty in simulation of BMRIS effect on the Southern Ocean. Besides, the effects of other ice shelves, such as the Filcher-Ronne and so on, should also be evaluated."

[C] p. 4 Table 1: Please reformat values in scientific notation rather than "e" notation used in programming languages (e.g. $1.0 \ 10^{-4}$ if you are using LaTex). Here and elsewhere, "m/s" should be "m s^{-1}" and similarly "m/a" should be "m a^{-1}", etc.
[A&R] Those values have been reformatted in scientific notation.

[C] p. 4 Table 1: Could you explain the choice to use ISOMIP thermodynamics? Neglecting the velocity dependence of the heat- and salt-transfer coefficients has been shown to reduce the accuracy of melt fields, see discussions in:
Dansereau V, Heimbach P, Losch M. Simulation of subice shelf melt rates in a general circulation model: velocity-dependent transfer and the role of friction. Journal of Geophysical Research: Oceans. 2014;119(3):1765–90. https://doi.org/10.1002/2013JC008846.
Asay-Davis, Xylar S., Nicolas C. Jourdain, and Yoshihiro Nakayama. "Developments in Simulating and Parameterizing Interactions Between the Southern Ocean and the Antarctic Ice Sheet." Current Climate Change Reports, October 24, 2017, 1–14.

https://doi.org/10.1007/s40641-017-0071-0.

[A] Although in most recent simulations models used have been updated from velocity-independent to dependent formulations, the impact has not been well documented (except for Pine Island Ice Shelf and Larsen C ice shelf). Especially the analysis for RIS could not be found. Under the current coarse resolution, I am not convinced the velocity-dependent formulation can improve the result significantly.

[R] A brief discussion has been added: "In the work the ISOMIP thermodynamics, which neglects the velocity dependence of the heat- and salt-transfer coefficients, has been used. In the velocity-independent melt rate parameterizations, the impact of currents or tides on the distribution of sub-ice shelf melting is indirect, hence limited (Dansereau et al.,2014). If the velocity dependence of transfer coefficients is considered, just as most recent modelling with fine grids did (Dansereau et al.,2014; Asay-Davis et al.,2017), differences in melt rate patterns may are found. The differences in melt rate patterns may be bigger in higher resolution modelling since high boundary layer currents can be resolved better. "

[C] p. 4 Table 1: Could you explain why the Jenkins et al. (2001) form was not used? They show that this can lead to a drift away from the expected linear relationship between T and S over long timescales, which seems problematic given that this study is focused precisely on long timescales.

[A] For ISOMIP thermodynamics, the salinity uses a conservative boundary condition that implicitly includes both advective and diffusive fluxes; the advection of percolating meltwater into the ocean, which having an impact on the ice-ocean heat flux, is generally small and could be overlooked.

[C] p. 4 l. 11-12: Could you please explain the choice to remove the ice shelf cavity in the 4 grid boxes rather than thicken the cavity? What criterion was used to decide whether the cavity is too thin and should be set to zero? How does the cavity thickness in the model compare with that of the original RTOPO-1 data set, averaged over each grid cell? Was the cavity thickness increased in some cells to match some required threshold (e.g. the column is more than x cells thick)? If so, was the ice draft moved up or was the bathymetry moved down, or both? What is the area of the modeled cavity compare to the area in RTOPO-1 and what would you expect the effect of this difference to be (I would expect the modeled cavity is much smaller and that this would lead to a reduced freshwater flux but a similar melt rate to observations). In summary, more explanation of the method is needed.

[A] For the 4 grid boxes whose ice shelf cavity are removed, the thicknesses of water columns are less than 42 m which can not be resolved with vertical grids of 50 m in size (starting from the 8th layer which is about 200 m below sea surface, the vertical grid size is 50 m) . If the cavity is too thin to be resolved by the vertical grids, it will be set to zero. The cavity thickness in the model is smaller compared to that in the original RTOPO-1 data set with the maximum difference less than 50 m. The cavity thickness was not increased in some cells to match some required threshold. In RTOPO-1the area of cavity is 502024.1 km^2 whereas in the model it is only 476924.2 km^2 due to the coarse model resolution.

[R] More explanation of the method has been added.

[C] p. 4 l. 12: I believe "depth" actually refers to "water-column thickness". Is that correct? If so, please make this substitution.
[A&R] That is correct. It has been revised.

[C] p. 4 Fig. 1: "indicate grids where cavities are resolved". Since 4 grid boxes have water-column thicknesses of zero, I would argue those grid boxes don't resolve the cavity and should probably be removed from the figure or shaded differently.
[A&R] It has been revised to "indicate grids which are covered by RIS in the model"

[C] p. 4 Fig. 1: I am deeply concerned that RIS, the main focus of the study, is captured by only 15 grid cells and with seemingly 50 m vertical resolution and seemingly without partial bottom cells (though neither of these are discussed in the text). The introduction suggests that it is important to capture the sub-ice-shelf flow in models because it cannot be observed directly, but such coarse resolution seems entirely inadequate to do that job.
[A] About 80% of RIS area is resolved by the model. For grid boxes with ice base exceeding 200 m below the sea surface, the vertical grid size is 50 m. The partial bottom cells are used in the model. It's true that the model cannot capture the sub-ice-shelf flow well. The aim of the work is not to simulate the sub-ice-shelf flow.

[C] p. 5 l. 9-13: It would be helpful to have a figure, panel of a figure or table to compare these various melt rates. It would be useful to be more quantitative than "larger" and "smaller". It would also be important to separate results derived from modeling from those derived from satellite measurements. It is encouraging that the melt rate lies within the range of observational and previous model estimates. What about freshwater fluxes (given that the area of RIS in the model is probably significantly different from observations)? How do these compare with other studies?
[A] The suggestions are accepted. Since the model can capture about 80% of RIS area, the influence of reduced area is not large considering the uncertainties of RIS melting in observations and modelling.
[R]    A table has been added in the revision version.
Table A2. Basal melt rates averaged over the entire RIS in the work and other studies

| Basal melt rates (m/a) | Source | Brief description |
| --- | --- | --- |
| 0.12 $\pm$ 0.03 | Shabtaie and Bentley (1987) | Calculated from the measured ice flux into the Ross Ice Shelf and previous measurements |
| 0.18-0.27 | Hellmer and Jacobs (1995) | Calculated from a two-dimensional (y/z plane) channel flow model forced by density differences between |

| | | |
|---|---|---|
| | | the open boundaries and the interior cavity |
| 0.25 | Assmann et al. (2003) | Calculated from a circumpolar numerical |
| 0.082 | Holland et al. (2003) | Calculated from a regional numerical model (MICOM) |
| 0.13-0.15 | Dinniman et al. (2007) | Calculated from a regional numerical model (ROMS) |
| 0.15 | Dinniman et al. (2011) | Calculated from the ROMS model |
| 0.6 | Timmermann et al. (2012) | Calculated from a global finite element ocean model (FESOM) |
| 0.0$\pm$ 0.1 for Ross West 0.3 $\pm$ 0.1 for Ross East | Rignot et al. (2013) | Calculated from radar measurements and output products from the Regional Atmospheric and Climate Model RACMO2 |
| 0.14 $\pm$ 0.05 | Depoorter et al. (2013) | Calculated from radar measurements and a regional climate model (for firn air content and compaction) |
| 0.25 (without tidal forcing) 0.32 (with tidal forcing) | Arzeno et al. (2014) | Calculated from the ROMS model |
| 0.11 $\pm$ 0.14 (converted from basal melt budget of RIS dM/dt in Table 3 with ice density 918 kg/m^3) | Moholdt et al. (2014) | derived from Lagrangian analysis of ICESat (NASA's Ice, Cloud and land Elevation Satellite) altimetry |
| 0.24 (converted from basal melt in Gt/yr for the last year of simulation in R_MLT | Mathiot et al. (2017) | Calculated from a regional numerical model (NEMO) |

| in Table 3 with RIS area 500 000 km2 and ice density 918 kg/m^3) | | |
| --- | --- | --- |
| 0.25 | This study | Calculated from quasi-equilibrium state of a global numerical modelling (MITgcm) |

[C] p. 5 l. 13: What is meant by "net melt rate"?

[A&R] In Holland et al. (2003), "net melt rate" refers to the sum of "melt-only" rate and "freeze-only" rate. It's identical to the basal melt rate in the work. To avoid confusing, it has been changed to "basal melt rate".

[C] p. 5 l. 15: What is meant by "model system evolution stage"? Does this refer to the numerical methods used to discretize the equations of motion?

[A] That means stage during the process of modeled ocean adjustment.

[C] p. 6 l. 7-8: "salinity bias and temperature bias": I don't think "bias" is the correct word here, as this would assume that the control case (without melting) are the observations, which they most certainly are not. I would also suggest avoiding the word anomaly unless you make clearer why you have chosen the EN experiment to be the "control" (implying you expect it to be the "normal" case in some sense). I think the most correct term, free from value judgments, would simply be "difference". So the sentence should probably read something like "The relationship between salinity and temperature differences in RIS cavity water between the two experiments..."

[A&R] The suggestion is accepted and the sentence has been revised.

[C] p. 6 l. 7-11: This linear relationship between T and S resulting from melting is well known and is called the Gade line:

Gade, HG. "Melting of Ice in Sea Water: A Primitive Model with Application to the Antarctic Ice Shelf and Icebergs." Journal of Physical Oceanography 9 (1979): 189 – 198.

It would be important to show if your line has the expected slope for a Gade line. Otherwise, it could indicate something is amiss with the sub-ice-shelf boundary conditions.

[A] This line reflects the relationship between difference of T and difference of S, it's not the Gade line. I haven't got the way to calculate characteristic parameter of the Gade line for water in the cavity from EI.

[C] p. 6 l. 10: "ppt" should probably be "PSU", which is slightly different. I do not believe MITgcm uses ppt to measure salinity.

[A&R] The practical salinity scale is used in the model. In some publications using MITgcm, the salinity is unit-less. I have revised the unit to PSU.

[C] p. 6 l. 11: "there seems to be no significant influence on the inflow and outflow in the cavity": The only way I can make sense of this phrase is if "influence on" is changed to "difference between". Melting clearly has an influence on both the inflow and the outflow so it is clearly not correct to say there is no influence. I would suggest that this finding deserves more discussion. The only way to make sense of this is that, in quasi-equilibrium, a significant amount of outflowing freshwater recirculates into the cavity. This is a somewhat surprising finding and I think possibly a significant difference between these simulations and those at higher resolution (e.g. Nakayama et al. 2014, Dinniman et al. 2017):

Nakayama, Y., R. Timmermann, C. B. Rodehacke, M. Schröder, and H. H. Hellmer. "Modeling the Spreading of Glacial Meltwater from the Amundsen and Bellingshausen Seas." Geophysical Research Letters 41, no. 22 (November 2014): 7942–49. https://doi.org/10.1002/2014GL061600.

[A] As suggested, it would be safer to change the phrase "influence on" to "difference between". Under the current resolution, the model cannot catch circulation in the cavity in detail.

[R] The phrase "influence on" has been changed to "difference between"

[C] p. 7 Fig 3: "ppt" should be "PSU". Typically, we use degrees C instead of K in cryospheric research but that makes no difference for this particular plot.

[A&R] Accepted and revised.

[C] p. 7 l. 6-20: Most of this paragraph seems simply to describe Fig. 4 without providing any physical insight into why these differences occur. To a limited degree, it is helpful to have you point out the most salient features of each panel but it would be far more useful to get some understanding of why changes in salinity occur where they do (and similarly for temperature). Why are they so different?

[A] I agree with you. To get a clear picture behind Fig. 4 is a hard work that I've tried for quite some time.

[R] See [R] parts of the following three comments.

[C] p. 7 l. 13-15: It is not at all obvious to me how you are backing up the assertion that freshwater flux is more significant than heat flux. The way I would expect to see that is in the influence of each on density changes, which in turn affect large-scale overturning and mixing into the deeper ocean. But Fig. 4 provides no information about the effects on density. Given that T and S have completely different units, there seems to be no basis for comparing the relative importance of these differences on their own. The fact that temperature differences are more scattered does not seem in any obvious way to support the conclusion that heat fluxes are less influential on these differences.

[A] In polar oceans, salinity has larger influence on density variation than temperature. I have added a brief discussion on that.

[R] A brief discussion is added: "By using the International Thermodynamic Equation of

Seawater-2010 (TEOS-10), the total differential in density $\rho$ can be expressed as

$$dρ = \frac{\partial ρ}{\partial \Theta}\big|_{S_A,p}\, d\Theta + \frac{\partial ρ}{\partial S_A}\big|_{\Theta,p}\, dS_A = ρ(-αd\Theta + βdS_A)$$

where $α = -\frac{1}{ρ}\frac{\partial ρ}{\partial \Theta}\big|_{S_A,p}$ , $β = \frac{1}{ρ}\frac{\partial ρ}{\partial S_A}\big|_{\Theta,p}$ ; $S_A$ is Absolute Salinity, $\Theta$ is Conservative

Temperature, $p$ is pressure, $α$ is the coefficient of thermal expansion ,and $β$ is the

coefficient of haline contraction. Using the Gibbs Seawater Oceanographic Toolbox in Fortran

(https://github.com/TEOS-10/GSW-Fortran), such variables as $S_A$, $\Theta$, $α$, $β$ and $ρ$ can

be easily computed. The $S_A$ difference-$\Theta$ difference distribution of water in the RIS cavity (EI

minus EN) is similar to Figure 3 (not shown). In polar oceans, $β$ is at least several times bigger

than $α$ (see Fig.S2). This implies that the change of density is more sensitive to that of
salinity. The added fresh water reduces the salinity in water body near the RIS. The reduced
salinity gives rise to a reduction of sea water density (Fig. S3).

[Figure]

Figure S2. Ratio of the coefficients of haline contraction and thermal expansion ( $β/α$ ) at 390 m

in EN. The units of $β$ and $α$ are kg g^-1 and $^\circ$C^-1 respectively.

[Figure]

Grid position in y direction

Figure S3. Difference of density (EI-EN) in the cross-section along x=351. The contour interval is 0.02 kg m^-3.

[C] p. 7 l. 19: I think it would make sense to include the figure indicated by "Figure not shown", as I think the changes in the ACC would be an important finding.
[A&R] The ACC is also reduced at bout 1000 m. The original analysis is not correct. The related sentences have been removed.

[C] p. 7 l. 19-20: The discussion of Fig. 5 is so short that it is not at all clear what the figure is justified. I did not get any physical insight into the spatial pattern of freshening at the seafloor from the figure or the discussion here.
[A&R] The figure has been redrawn. Added discussion: The BMRIS has the biggest influence on bottom water in the Southern Pacific Ocean, especially the Ross Sea and its adjacent western (looking from the north) deep ocean. The signal of the BMRIS effect is weak in the Southern Atlantic Ocean compared to those in the Southern Pacific Ocean and the Southern Indian Ocean. This result agrees with the picture of the thermohaline circulation, in which the deep current moves southward in the Atlantic Ocean."

[Figure]

Fig. 5

[C] p. 8 Fig 4: The axis need descriptive labels including units. Tick mark labels should be larger. Caption should include the color of the curves (since figure will always be in color). There is no obvious reason that the x axes of the 3 panels are different, and this makes comparing the panels more difficult. The x axis is for both salinity and temperature differences? The depth axis should be inverted so that the deep ocean is down. It is also standard to have these depths be negative, indicating that they are elevations below sea level. What is the northern boundary of each of these regions? What longitudes separate them?

[A&R] The x axis is for both salinity and temperature differences. The northern boundary of each of these regions is 35 ⁰S; the Southern Indian Ocean is from 19 ⁰E to 145 ⁰E; the Southern Pacific Ocean is from 146 ⁰E to 290 ⁰E; the Southern Atlantic Ocean is from 69 ⁰W to 18 ⁰E (Fig. S1). Fig. 4 has been redrawn.

[Figure]

Figure S1 Division of world ocean in 1 x 1 longitude-latitude grids

[Figure]

Fig. 4

[C] p. 9 Fig 5: This figure does not seem at all useful to me. The color contours are set such that all we can tell is the sign of the salinity difference (and that it is greater than -0.05 PSU) over the vast majority of the sea floor. A nonlinear color bar or one with many more contour values would be needed to make this figure at all useful.
[A&R] The figure has been redrawn (see my previous answer).

[C] p. 9 l. 5 "surface ocean": A careful point has been made in the manuscript that the freshwater flux is not at the ocean surface, so this should probably be "upper ocean".
[A&R] the phrase "surface ocean" has been revised to "upper ocean".

[C] p. 9 l. 5-p. 10 l. 2: This paragraph again refers to "anomalies", whereas I would encourage you to use "differences". Other than this small issue, I think this paragraph has some of the best analysis in the paper.
[A&R] I agree with your feeling about the difference between "anomaly" and "difference". But in

some cases, it is inappropriate to use "difference" instead of "anomaly". For example, cold anomaly makes sense but cold difference does not, right? In cases "anomaly" can be replace with "difference", I have made substitution as far as possible.

[C] p. 9 l. 9-10: It's not clear to me what the difference between the warm advection anomaly and the warm SST anomaly is. It seems obvious that the one would cause the
other but maybe I'm missing something.
[A] The advection involves flow field. The former can lead to the latter, but the latter cannot lead to the former without favorable flow condition.

[C] p. 9 l. 10-11: "The cold water from BMR is advected by the ACC westward": A couple of things here, the ACC flows eastward (which seems to be the direction most of the cold difference is being advected) not westward. There is also the Antarctic Coastal Current (ACoC) that does flow westward on the continental shelf so maybe that's what is advecting a bit of the colder melt water to the west toward that SIT dipole?
[A&R] I used a wrong word. It should be "eastward" (looking from the South Pole. I am looking from North). I have changed the "westward" to "clockwise". Maybe due to the coarse resolution, the model cannot reproduce the Antarctic Coastal Current (ACoC) well.

[C] p. 9 l. 15: I don't understand the cause of the increased SST near the sea-ice edge. Could you explain further why downwelling is associated with increased SST?
[A&R] The initial explanation is not accurate. The sentence has been revised to "This gives rise to piling up of warm water and increasing of SST in EI "

[C] p. 9 l. 16: It seems worth exploring in more detail *how* the results from the two studies are different, not just to point out that they are different and that they are simulating different conditions (transient vs. quasi-steady; cavity geometry vs. no cavities for the
"control").
[A] The suggestion is accepted.
[R] revised: The feature of SIT difference in this work is quite different from that of Hellmer (2004), in which SIT in the Ross Sea gets thicker and there is no significant difference in SIT in the ocean area downstream the Ross Sea. In his work, the result of the 20th model year from a regional coupled ice-ocean model is given and the RIS cavity geometry is not included in the model bathymetry for the no sub-ice freshwater input experiment. Perhaps the differences in results between the two works are at least to a great deal due to the different treatments for the RIS cavity geometry in the no sub-ice melting experiments.

[C] p. 10 Fig 6: labels (tick marks, lat/lon, color bars) are all far too small. Please make them bigger and crisper. Please add more lines for lat and lon if possible so the reader can more easily find the lat/lon coordinates identified in the text.
[A&R] The figure has been redrawn.

[Figure]

[C] p. 11 l. 9-10: It seems entirely backwards to me that including basal melting would decrease the MOC. If the model were correctly producing more AABW from ice-shelf melting and subsequent climate and topographic interactions, there should be an increase in downwelling just off the Ross continental shelf break and an associated increase in southward transport in the upper ocean (by conservation of mass). This should lead to an increased MOC strength. This is my understand of the main contribution of Antarctic climate dynamics to the global ocean circulation. To me, the decreased MOC in your simulations with melt fluxes suggest that something is wrong in the simulations and AABW is not being produced. This would not be surprising at coarse resolution, since ESMs have a very hard time producing AABW for the right reasons at CMIP-type resolutions.

[A] I do not know what backwards mean here. Does it mean the result here has been proved to be out-of-date or wrong? Or does it mean the result is opposite to what you expect? My result supports studies such as Hellmer (2004) and Kusahara and Hasumi (2013), whose model resolutions are not coarse and their integrations are short compared to this work.

[C] p. 11 l. 13: The formation and spreading of AABW should be the cause (not the effect) here. Changes in AABW formation should be driving the changes in the MOC.
[A] I agree with you. The BMRIS influences AABW, which influences MOC subsequently. The results agree with the idea.

[C] p. 12 l. 4-6: I suggest you look further into these difference as part of this paper. It is precisely this kind of comparison with previous work that I feel is missing from this paper. Without more of this kind of validation work, it remains hard to trust the conclusions about the effects of melt fluxes on the ocean-sea ice system.
[A] The suggestion is accepted.
[A] addition: The strength and position of Subpolar Cell, Upper Cell and Lower Cell in this model resemble those in ACCESS and GFDL-MOM given in Farneti et al. (2015) much more. The strength and position of simulated Cells given in Farneti et al. (2015) are varied. The biggest discrepancy among the models exists in the strength of the anti-clockwise Lower Cell, which ranges from 20 Sv to zero. The simulated strength of the Lower Cell from EI is about 15 Sv (Fig. 9).

[C] p. 12 Fig. 9: I find it very hard to tell what is going on with the difference contours. The color plot is quite clear in most regions but hard to discern near the Antarctic the contours are hard to get the sign of, let alone the magnitude in the Antarctic. Maybe the figure should give more space to the region from -90 to -60 (i.e. a nonlinear x axis).

[A&R] The figure has been revised.

[Figure]

Fig. 9

[C] p. 12 l. 14: "contributes to northward heat transport anomaly": I find this confusing, since at least in the real world there should be a consistent southward transport of heat. In your simulations, you seem to see a mix of northward and southward transport do the "anomaly" is contributing to a reduction in southward heat transport at some latitudes and enhanced northward transport in others. Maybe "contributes to a reduction in southward heat transport"? Also, this needs some discussion. Consistent with my concern about the MOC above, it seems like you should be seeing steady southward heat transport in both cases and that southward heat transport should be enhanced by AABW formation, whereas you are seeing a consistent global reduction (with varying behavior in each ocean basin). The discussion of the individual basins is clearer in terms of describing enhanced or reduced transport.

[A] Yes, the "anomaly" is contributing to a reduction in southward heat transport at some latitudes and enhanced northward transport in others. I recalculate the heat transport with monthly averaged VT instead of V and T, the northward transport in the Southern Ocean vanishes. In the simulation result the AABW formation is reduced if the effect of BMRIS in included.

[R] The phrase "enhanced" or "reduced" is used instead of "anomaly" in situations describing transport change.

[C] p. 13 Fig 10: It would be helpful to compare the global MHT in 10a with observations, such as: Trenberth, Kevin E., and Julie M. Caron. "Estimates of Meridional Atmosphere and Ocean Heat Transports." Journal of Climate 14, no. 16 (August 1, 2001): 3433–43. https://doi.org/10.1175/1520-0442(2001)014<3433:EOMAAO>2.0.CO;2.

Eyeballing the comparison, the global MHT isn't too bad north of 40S but it is odd that you are seeing significant *northward* transport of heat between 60S and 40S, which is not consistent

with observations.

[A] Thanks for giving the article. By using model output of monthly averaged VT directly instead of V and T, the calculated northward transport in the Southern Ocean vanishes and agrees with that of Trenberth and Caron (2001) better. Compared to Trenberth and Caron (2001), the curves of heat transport are not smooth and slopes in some latitudes are large for the individual basins. Since the heat transport for the individual basins are less reliable, analysis on them will be removed in the revision.

[R] Analysis on the heat transport for the individual basins have been removed. Analysis on comparing the global MHT with Trenberth and Caron (2001) has been added:
"

[C] p. 13 Conclusion and discussion: Overall, I find that this is mostly just a summary of the results with insufficient interpretation of the findings, discussion of the implications of this work for other modeling efforts and/or the behavior of the "real world" and insufficient introspection about what the missing processes and other shortcomings of the work might be.

[A] Yes, the initial conclusion and discussion needs improving.

[R] Some discussions have been added as given in previous [R] parts.

[C] p. 13 l. 9: "profoundly": This is a very subjective term and I'm not sure it is supported by the results. The differences between simulations with and without RIS melting are detectable to be sure but the changes generally seem to be subtle rather than profound.

[A&R] The word is removed.

[C] p. 13 l. 9-p. 14 l. 2: My concerns about the "latent heat flux anomaly" and associated complexity of the temperature evolution remain the same as above. I do not think there has been sufficient analysis of the physical processes leading to the temperature evolution to conclude that they are even the result of the latent heat flux from ice-shelf melting. Instead, they are likely to result primarily from density changes, which are in turn primarily controlled by freshwater fluxes. Thus, I think the conclusion that the latent heat flux plays a secondary role is correct but I don't think anything presented in this manuscript has supported that conclusion directly.

[A] I agree with you. In previous [R] part I have added some analysis on the influence of temperature and salinity on density.

[C] p. 14 l. 3-9: The manuscript did not present the circulation from either EI or EN or make any attempts to compare these with observations, so it is difficult to know how much (if any) credence can be given to the difference in circulation between the two experiments. That being said, Again I find the discussion of the surface processes to be among the most useful analysis in the paper.

[A] There are totally 91 boxes in the cavity. The current configuration cannot resolve circulation in the cavity in detail and there could be no favorable things to share.

[C] p. 14 l. 7: Again, the fact that basal melting stabilizes the water column and weakens overturning just seems to indicate that the processes we know to occur as part of AABW formation are missing from the model.

[A] Since the vertical resolution is coarse near the sea bottom, it is more possible that AABW formation is not depicted well than other models with finer resolution. But there are other models whose resolutions I believe are fine enough also give similar results.

[C] p. 14 l. 10-13: The discussion of fixed ice-shelf area seems unrelated to the manuscript and its findings. There is nothing to suggest that having dynamic ice-sheet geometry in this configuration would enhance our understanding of the quasi-equilibrium state of the ice sheet-ocean-sea ice system because: 1) the resolution of the ocean model is very much insufficient to supply realistic melt patterns to drive ice-sheet evolution; 2) the steady-state melting, if consistent with present-day observed melting, would be unlikely to drive any significant ice-sheet evolution because melt rates under RIS are very small. 3) the context in which melt-driven ice sheet dynamics are interesting are precisely those that are *not* in quasi-equilibrium.
p. 14 l. 14-19: I appreciated this discussion of possible future directions for the research.
[A&R] The suggestion is accepted and the paragraph on discussion of fixed ice-shelf area has been removed in the revised version.

[C] Typographical and Grammatical Corrections:
Title: The title would read better as "Modelling the effect of Ross Ice Shelf melting on the Southern Ocean in quasi-equilibrium"
[A&R] Accepted.

[C] p. 1 l. 6: "basal melting of Ross Ice Shelf" should be "basal melting of *the* Ross Ice Shelf"
p. 1 l. 19: remove "And, ". It is not necessary and is grammatically incorrect.
[A&R] Corrected.

[C] p. 1 l. 20: "accompanied accordingly": This phrase doesn't make sense. Perhaps you mean something like, "There is an accompanying northward anomaly in meridional heat transport at most latitudes of the global ocean"?
[A&R] Yes, that is what I want to express. It has been corrected.

[C] p. 1 l. 23: "Ices accumulated... are" should be "Ice accumulated ... is". Ice is only plural if there are multiple classes of ice or something along those lines, which doesn't seem to be the case here.
p. 2 l. 11: 2 should be written out a "two".
p. 2. l. 13: I suggest changing "regarding" to "of".
[C] p. 2 l. 17: here and elsewhere "sub-ice shelf" should be "sub-ice-shelf"
p. 2 l. 19: "representation" should be "representations"
p. 2 l. 21: "parameterization should be "parameterized"
[A&R] Corrected.

[C] p. 2 l. 27: "In study such as modeling Ice Shelf melting effect on the Ocean…" this whole sentence is needs some significant grammatical work. Here's my best guess at

what is intended: "In studies that include the effect of ice-shelf melting on the ocean in quasi-equilibrium, it is necessary to use a global model with thermodynamically active ice-shelf cavities and to perform integration over hundreds of years"

[A&R] Your guess is correct. It has been corrected.

[C] p. 3 l. 13-14: "assuming the RIS being in steady state" should be "assuming the RIS to be in steady state"

p. 3 l. 17: "should be "to get *the* RIS draft"

p. 3 l. 29 and 31: "1ooo" should be "1000" (zeros, not o's).

p. 3 l. 29: "To resolve the RIS vertically better" should be "To better vertically resolve the RIS"

p. 4 l. 3: "to that in" should be "to those in"

p. 4 l. 8-9: "and the Antarctic situates on the.." should be "with Antarctica situated on the..."

p. 4 l. 9: "the bathymetry of ocean around the Antarctica and cavity geometry of RIS is" should be "the ocean bathymetry around Antarctica and the cavity geometry of RIS are"

p. 4 l. 10: "grids" should be "grid cells" or "grid boxes". To me, the whole 64x64 face is a grid.

p. 4 l. 11: "of which 15 having cavities and being calculated basal melting" should be something like "of which 15 have nonzero cavity thickness and include basal melt calculations"

[A&R] Corrected.

[C] p. 4 Fig. 1: "(a)" and "(b)" should go before the phrases describing each panel rather than after.

[A&R] Corrected.

[C] p. 4 Fig. 1: "yellow shades in (b) indicate grids where cavities..." should probably be "grid boxes shaded light green indicate locations where cavities...". (To my eyes, the shading is light green, not yellow.)

[A&R] I used a wrong word. It has been corrected. Thanks.

[C] p. 5 l. 15: "modelling ice shelf" should be "modeled ice shelf". "lateral boundary" should be "lateral boundaries".

[A&R] Corrected.

[C] p. 5 Fig 2: The tick mark labels on the axes are too small to easily read. The melt-rate values are also somewhat small but perhaps large enough to read (but I see no reason to include so many empty cells around the 15 active cells. The 3 panels probably will need to be combined into a single figure for typesetting but I guess that's up to you and the journal to work out.

[A&R] The figure has been redrawn.

[Figure]

[Figure]

[Figure]

Fig. 2

[C] p. 5 Fig 2: "annual mean areal average" should probably be something like "the annual and area mean"; "for the last 100 years' mean" should be "for the mean over the last 100 years"; "areal mean averaged over the last 100 years" might be clearer as "averaged over the ice-shelf area and the last 100 years".

p. 6 l. 6: "cold and fresh water are" should be "cold and fresh water *is*"

p. 6 l. 7: "become" should be "becomes"

p. 6 l. 7: "compared its counterpart" should be "compared *to* its counterpart"

[A&R] Corrected.

[C] p. 6 Fig 3: The axis labels should be more descriptive (not variable names) and should include units.

[A&R] The figure has been redrawn.

[Figure]

Fig. 3

[C] p. 7 l. 1: "Figure 3. Figure 3." should just be "Figure 3"

p. 7 l. 6-20: There is no need to continually reference Fig. 4 here. It is clear that most of this text refers to that figure.

p. 7 l. 7: "from ocean surface" should be "from *the* ocean surface"

p. 7 l. 9: "freshening effect" should be "*the* freshening effect"

[A&R] Corrected.

[C] p. 7 l. 16: why 1005 m instead of just 1000 m?

[A&R] The model layer situates at 1005 m. The phrase "1005 m" has been revised to "about 1000 m"

[C] p. 7 l. 17-18: "This is due to that": this phrase is kind of confusing. I would suggest something like "This is due to the relatively stronger...at that level, which constrains..."

[A&R] It has been revised as suggested.

[C] p. 7 l. 19: "water in most area" should be "water in most of the area"

p. 9 l. 16 and p. 10 l. 1: "the work" should be "this work"

[A&R] Corrected.

[C] p. 10 l. 12: "BMR effect" should be "the BMR effect" (or maybe "the Ross BM effect", see earlier comment).

[A&R] It has been revised to "the BMRIS effect" (see earlier answer).

[C] p. 11 l. 1: No need to reference Fig. 8 again.

p. 11 l. 2: "Figures not shown" should just be "not shown"

p. 11 l. 3: "motion field" should be "flow field"

p. 11 l. 8: "by meridional transport" should be "by a meridional transport" or "by the meridional transport"

p. 11 l. 10 "here" should be lowercase or this should be made a separate sentence in parentheses (though The Cryosphere's typographic editors discourage theses)

p. 11 l. 15 "the path" should be lowercase

p. 11 l. 15 "it's" should be "it is"

p. 12 l. 2 "the calculation" should be lowercase

[A&R] Corrected.

[C] p. 12 Fig. 9: Axes need labels including units. All labels are too small to be readable.

[A&R] The figure has been redrawn. See previous answer.

[C] p. 12 l. 14: "contributes to northward" should be "contributes to the northward"

[A&R] Corrected.

[C] p. 13 Fig 10: the customary way of handling multiple y axes is to put one axis on the left of the figure and the other on the right. It is even more helpful if the axes are the same color as the curves they correspond with. As in all figures, the tick mark labels are far too small.

[A&R] The figure has been redrawn.

[Figure]

Fig. 10

[C] p. 14 l. 7: "stables" should be "stabilizes"

[A&R] Corrected.

---

## Referee Report (RR1)

Review of revised version of "Modeling the effect of Ross Ice Shelf melting on the Southern Ocean in quasi-equilibrium" by Xiying Liu.

Ryan Walker
8 August 2018

The revised version has been greatly improved from the original, and is now very clear and much more interesting. I thus have only a few suggestions to offer.

I originally commented about the (at least to me) unusual choice to compare model experiments with full sub-ice shelf thermodynamics and bathymetry to a scenario with bathymetry but no sub-ice shelf melting instead of to a scenario where ice shelves are entirely excluded.

Your response is very interesting, as I generally work on understanding the interplay between ocean thermodynamics and ice shelf/stream dynamics, but haven't thought too much about the cavities influencing large-scale ocean dynamics solely through their geometry/bathymetry. I haven't seen any other modeling that attempts to separate thermodynamic effects from purely bathymetric effects, and I think this idea is a good contribution, especially given what you've said about the bathymetry as a larger influence on the circulation. Comparison between two effects that are real can be more interesting than comparison with a scenario (no ice shelves) that is a model deficiency rather than a physical process.

However, I don't see this explanation written in the revised manuscript. I think a paragraph (probably in section 2) outlining the reasons for your choice of experiments to present would be useful to many or most readers. It would be beyond the scope of this short, well-focused paper to also present your experiments with ice shelves excluded. But a few comments (and maybe a supplemental figure if you want) regarding your preliminary work should help to clarify things.

Other comments:

Section 1, Line 14) Cut "The" at start of sentence.
Section 1, Line 17) Better to write the melt rate in meters per year for consistency with the rest of the manuscript.
Section 3.1/Figure 2) Could you comment on the spatial distribution of the seasonal cycle in your experiments, and what the likely causes are? From earlier work that I'm familiar with, I'd expect that it would be changes in warm currents near the ice front, but I'm curious what you got.
Figure 5) This color scale is much easier to read than the original, but maybe point out in the caption that the intervals are uneven.
Section 3.2, Line 5) Be clear that you're linearizing the equation of state here.
Figure 7) I think subplot (a) needs its own color bar.

---

## Author Response (AR2)

I want to thank R. Walker for his constructive review and good suggestions. I am answering his comments in the following. For clarity, I repeat the original comment ([C]) at first and then the answer ([A]) and author's changes in manuscript [R] afterwards: General Comments

[C]: Inclusion of ice shelves in global circulation models is a significant issue for the accuracy of climate projections. This study considers the impact of basal melting under the Ross Ice Shelf on the Southern Ocean by contrasting global ocean model experiments with and without melting in the sub-Ross cavity. The choice of a no-melt scenario that includes sub-ice-shelf bathymetry seems a little odd to me, as most ocean modeling that I'm aware of either includes ice shelves plus melting or excludes ice shelves from the domain. It should still be possible to get value from this experimental setup. However, I would have liked this manuscript to spend much more time on detailed discussion of the different experiments, particularly the relations between water properties and dynamics.

[A]: Initially I set up two experiments, one included ice shelves plus melting and the other excluded ice shelves from the domain. After preliminary analysis of simulation results, I realized that the sub-ice-shelf bathymetry gave significant contribution to the differences between the results from the two simulations. This difference in geometry changes local circulation and mixing and leads to changes of overall results compared to or even greater than that in basal melting under the Ross Ice Shelf. Under such conditions, it would be difficult to discuss the effect of basal melting under the Ross Ice Shelf. Hence a third experiment with no-melt scenario that includes sub-ice-shelf bathymetry was added and its results were used in the discussion instead of that from the experiment that excluded ice shelves from the domain. More discussions on the modeling results have been added. [R] See [R] parts for specific comments.

[C]: General comment on figures) All units should be in axis labels, not only in the captions. Also, axes should be labeled with variable names. Figures 1, 2, 6, 7, 9, 10 should have a larger

[A&R] These figures have been redrawn:

font size to be readable.

Fig. 1. (a) Bathymetry of the 6th cubed sphere face in the experiments and (b) cavity geometry of RIS in EI. The numbers on the axes indicate the positions of grids on the model domain.

Grid boxes shaded light green in (b) indicate the locations covered by RIS in the model and the numbers in (b) indicate the thickness of the water column in the cavity. The units of bathymetry and water column thickness in the cavity are in m.

---

## Author Response (AR3)

I want to thank R. Walker for his constructive review and good suggestions. I am answering his comments in the following. For clarity, I repeat the original comment ([C]) at first and then the answer ([A]) and author's changes in manuscript [R] afterwards:

General Comments

[C] The revised version has been greatly improved from the original, and is now very clear and much more interesting. I thus have only a few suggestions to offer.

I originally commented about the (at least to me) unusual choice to compare model experiments with full sub-ice shelf thermodynamics and bathymetry to a scenario with bathymetry but no sub-ice shelf melting instead of to a scenario where ice shelves are entirely excluded.

Your response is very interesting, as I generally work on understanding the interplay between ocean thermodynamics and ice shelf/stream dynamics, but haven't thought too much about the cavities influencing large-scale ocean dynamics solely through their geometry/bathymetry. I haven't seen any other modeling that attempts to separate thermodynamic effects from purely bathymetric effects, and I think this idea is a good contribution, especially given what you've said about the bathymetry as a larger influence on the circulation. Comparison between two effects that are real can be more interesting than comparison with a scenario (no ice shelves) that is a model deficiency rather than a physical process.

However, I don't see this explanation written in the revised manuscript. I think a paragraph (probably in section 2) outlining the reasons for your choice of experiments to present would be useful to many or most readers. It would be beyond the scope of this short, well-focused paper to also present your experiments with ice shelves excluded. But a few comments (and maybe a supplemental figure if you want) regarding your preliminary work should help to clarify things.

[A] The suggestion has been accepted and a few comments have been added in Section 2.

[R] In the model configuration part of section2, a few lines have been added:"An experiment with RIS excluded from the domain, denoted as EX, is also implemented. The experiment shares the configuration of EN except that the area covered with RIS is treated as land. After preliminary analysis of simulation results, it is realized that the sub-ice-shelf bathymetry gives significant contribution to the differences between the results from EX and EN. The difference in geometry changes local circulation and mixing and leads to changes of overall results compared to or even greater than that in basal melting under the RIS. Under such conditions, it would be difficult to discern the effect of basal melting under the Ross Ice Shelf. Hence the results from EX will not be discussed in later analysis."

Other comments:

[C]  Section 1, Line 14) Cut "The" at start of sentence.
      Section 1, Line 17) Better to write the melt rate in meters per year for consistency with the rest of the manuscript.
[A&R] Corrected.

[C] Section 3.1/Figure 2) Could you comment on the spatial distribution of the seasonal cycle in your

experiments, and what the likely causes are? From earlier work that I'm familiar with, I'd expect that it would be changes in warm currents near the ice front, but I'm curious what you got.

[A] The seasonal cycles for different grid cells show different features(Fig. A1). For grids adjacent to the ground and far from the ice front, the basal melt is influenced by flux from vertical heat conduction to a greater extent and reflects the air temperature variations much more.

[Figure]

Fig. A1 Seasonal cycle averaged over the last 100 years for different grid cells illustrated in Fig. 2b.

[R] 2 sentences have been added in the corresponding part: "It should be noted that the phase of seasonal cycle of basal melt is not uniform geographically. For grids adjacent to the ground and far from the ice front, the basal melt is influenced by flux from vertical heat conduction to a greater extent and reflects the air temperature variations much more (figures not shown)."

[C] Figure 5) This color scale is much easier to read than the original, but maybe point out in the caption that the intervals are uneven.

[A&R] The suggestion has been accepted and it has been noted in the caption.

[C]Section 3.2, Line 5) Be clear that you're linearizing the equation of state here. Figure 7) I think subplot (a)

needs its own color bar.

[A] Yes, it is a kind of linear approximation. I don't think the nonlinear effect can change the qualitative relations. With the same color bar in Fig.7 , the effect of **BMRIS** on seasonal variation of SIT is clearer.

I want to thank X. Asay-Davis for his constructive review and good suggestions. I am answering his comments in the following. For clarity, I repeat the original comment ([C]) at first, then the answer ([A]) afterwards:

General Comments

[C] Thank you for the significant revisions to the manuscript, and in particular for the more detailed analysis of model results, comparison with observations and caveats in the discussion section.

Thank you as well for the discussion in your response on the changes in the MOC. I now realize that my intuition may be mistaken and that the addition of ice-shelf melting may actually lead to a reduced AABW production and decreased global circulation. We may see that modeling at higher resolution may show this to be an artifact of unresolved processes, but your findings are at least consistent with other published work.

[A] Thank you so much for your previous thorough review which helps me a lot in improving the manuscript. I agree with you that the impact of model resolution on modeling results deserve further studies. If possible, I'll do some work on that later.

[revised manuscript text omitted]